# Styl3R: Instant 3D Stylized Reconstruction for Arbitrary Scenes and Styles

**Peng Wang**[1,2*]   **Xiang Liu**[2*]   **Peidong Liu**[2†]

[1] Zhejiang University    [2] Westlake University

{wangpeng,liupeidong}@westlake.edu.cn, liuxiangnick@gmail.com

**Project Page**

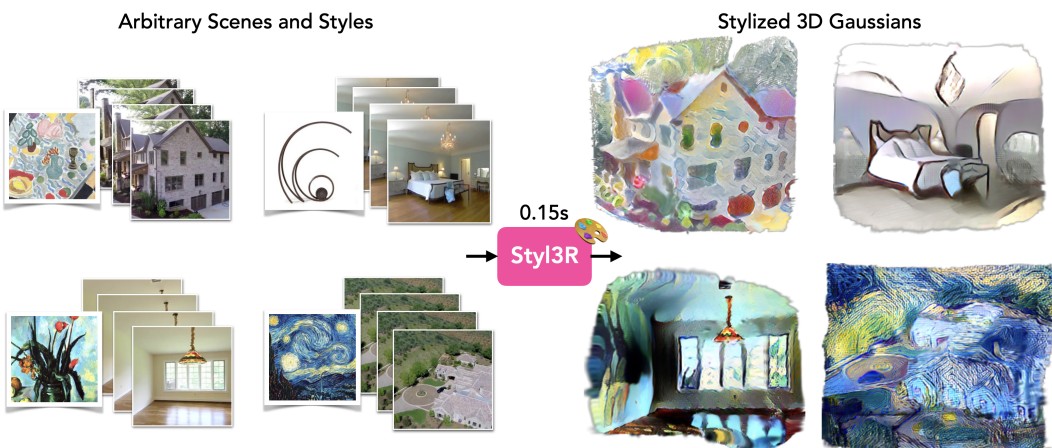

Figure 1: **Styl3R.** Given unposed sparse-view images and an arbitrary style image, our method predicts stylized 3D Gaussians in less than a second using a feed-forward network.

## Abstract

Stylizing 3D scenes instantly while maintaining multi-view consistency and faithfully resembling a style image remains a significant challenge. Current state-of-the-art 3D stylization methods typically involve computationally intensive test-time optimization to transfer artistic features into a pretrained 3D representation, often requiring dense posed input images. In contrast, leveraging recent advances in feed-forward reconstruction models, we demonstrate a novel approach to achieve direct 3D stylization in less than a second using unposed sparse-view scene images and an arbitrary style image. To address the inherent decoupling between reconstruction and stylization, we introduce a branched architecture that separates structure modeling and appearance shading, effectively preventing stylistic transfer from distorting the underlying 3D scene structure. Furthermore, we adapt an identity loss to facilitate pre-training our stylization model through the novel view synthesis task. This strategy also allows our model to retain its original reconstruction capabilities while being fine-tuned for stylization. Comprehensive evaluations, using both in-domain and out-of-domain datasets, demonstrate that our approach produces high-quality stylized 3D content that achieve a superior blend of style and scene appearance, while also outperforming existing methods in terms of multi-view consistency and efficiency.

---

[*]Equal Contribution;    [†] Corresponding author.

39th Conference on Neural Information Processing Systems (NeurIPS 2025).

# 1 Introduction

Recent advances in 3D reconstruction from 2D images [15, 27, 49] have made it increasingly practical to generate high-quality 3D scenes from casual captures. However, integrating artistic features such as the visual style of a reference image into these reconstructions remains technically challenging. This difficulty arises from the fundamental mismatch between style transfer and 3D reconstruction: altering the visual appearance to reflect a desired style often conflicts with the need to preserve multi-view consistency and structural coherence in the reconstructed scene.

Creating stylized 3D scenes typically requires professional expertise, artistic creativity, and substantial manual effort. While 2D image style transfer [9, 12, 20] has achieved impressive efficiency and visual fidelity, naïvely extending these techniques to 3D often results in view-inconsistent stylization. In contrast, recent 3D stylization methods [8, 23, 24, 54] offer improved appearance consistency but rely on dense multi-view inputs, known camera poses, and time-consuming per-scene or per-style optimization. These requirements render them impractical for casual users or time constrained applications. This leads to a central question: *How can we achieve fast, multi-view consistent 3D stylization from a few unposed content images and an arbitrary style image — without requiring test-time optimization?*

To address this challenge, we propose Styl3R, a feed-forward network that jointly reconstructs and stylizes 3D scenes from sparse, unposed content images and an arbitrary style image. Our method flexibly handles 2 to 8 input content images, requiring no test-time optimization, no camera supervision, and no scene- or style-specific fine-tuning, making it both practical and accessible for real-world usages. For clarity, we summarize the key differences between prior methods and ours in Table 1.

In particular, Styl3R adopts a dual-branch architecture consisting of a structure branch and an appearance branch. The structure branch predicts the structural parameters of 3D Gaussians from unposed content images by leveraging a dense geometry prior. The appearance branch, responsible for determining the color of the 3D Gaussians, consists of transformer decoder layers that blend style features with content features from multiple viewpoints. This design enables the generation of photorealistic and stylized appearances while maintaining multi-view consistency. To further preserve photometric shading capabilities during stylization fine-tuning, we introduce an identity loss by randomly feeding a content image into the appearance branch, encouraging the model to retain its ability to reconstruct the original appearance.

We evaluate our method on both in-domain and out-of-domain datasets. It outperforms prior approaches in terms of both multi-view consistency and efficiency, which producing high-quality 3D stylized content in only 0.15 second.

Our main contributions are:

- We introduce a feed-forward network for 3D stylization that operates on sparse, unposed content images and an arbitrary style image, does not require test-time optimization, and generalizes well to out-of-domain inputs — making it practical for interactive applications.
- We design a dual-branch network architecture that decouples appearance and structure modeling, effectively enhancing the joint learning of novel view synthesis and 3D stylization.
- Our method achieves state-of-the-art zero-shot 3D stylization performance, surpassing existing zero-shot methods and approximate the efficacy of style-specific optimization techniques, as demonstrated through both quantitative metrics and qualitative results.

# 2 Related Work

**2D Style Transfer.** The task of transferring the visual style of a reference image onto another content image has been extensively studied. The seminal work [9] introduced neural style transfer by iteratively optimizing the stylized image to match the Gram matrices of VGG-extracted features [36] from both content and style images. Subsequent approaches, such as AdaIN [12], WCT [21], LST [20], and SANet [31], reformulated the problem into a feed-forward setting by aligning the feature statistics of content and style. With the emergence of transformer-based architectures [6, 42], more recent methods like AdaAttN [25], StyleFormer [47], StyTr2 [5], S2WAT [52], and Master [40]

| | Method | Sparse View | Scene Zero-shot | Style Zero-shot | View Consistency | Pose Free | Fast Inference |
|---|---|---|---|---|---|---|---|
| 2D | Methods [5, 12, 25] | - | ✓ | ✓ | ✗ | - | ✓ |
| 3D | StyleRF [23] | ✗ | ✗ | ✓ | ✓ | ✗ | ✗ |
| | StyleGaussian [24] | ✗ | ✗ | ✓ | ✓ | ✗ | ✗ |
| | ARF [24] | ✗ | ✗ | ✗ | ✓ | ✗ | ✗ |
| | **Styl3R (Ours)** | ✓ | ✓ | ✓ | ✓ | ✓ | ✓ |

Table 1: **Comparison with existing style transfer methods.** 2D methods can instantly stylize images after training without any additional tuning, but they fail to ensure multi-view consistency. Prior 3D methods ensure consistency but rely on dense posed inputs and per-scene or per-style tuning, slowing inference. Our method combines both strengths, achieving view-consistent stylization in under a second without further tuning.

leverage attention mechanisms to enhance feature representation, moving beyond the limitations of intermediate features extracted from pretrained VGG networks [36]. Despite progress in visual quality and efficiency, 2D style transfer methods lack geometric awareness and multi-view consistency, often causing artifacts when naïvely extended to 3D. In contrast, our method achieves geometry-consistent, view-coherent stylization with fast inference and rendering.

**3D Style Transfer.** 3D stylization has been explored using various scene representations. Early methods leveraged explicit forms like meshes [10] and point clouds [11, 17, 28], enabling style transfer via differentiable rendering and geometric warping, but they struggled with complex scenes due to limited expressiveness. Recent work adopts implicit representations like NeRF [27] and 3D Gaussian Splatting (3DGS)[15]. StylizedNeRF[13] incorporates a 2D stylizer with view-consistent losses; ARF [54] improves structural fidelity with feature matching; StyleRF [23] enables zero-shot stylization via feature-space transformation; INS [7] disentangles style and content with geometric regularization. For 3DGS, StyleGaussian [24] and [34] embed style features into Gaussian parameters; SGSST [8] introduces multi-scale losses for high-res output; InstantStyleGaussian [51] stylizes via dataset updates; StylizedGS [53] achieves controllable editing by disentangling content, style, and structure. However, most methods require dense views, known poses, and per-scene optimization, limiting their applicability in casual settings. Other recent efforts [30, 38] explore object-centric stylization using attention and diffusion models (e.g. Zero123++ [35]) but suffer from high inference times ($\sim$30s per object). In contrast, our method does not directly rely on the output of external pretrained models and achieves stylization in under one second.

**Feed Forward Generalizable 3D Reconstruction.** Though NeRF [27] and 3DGS [15] achieve high-quality view synthesis, they rely on dense, calibrated inputs and costly per-scene optimization, hindering their practical usages for time-constrained applications. To address this, generalizable reconstruction methods [3, 14, 44, 48, 50] have emerged, primarily leveraging geometric priors like cost volumes or epipolar constraints to aggregate multi-view information from sparse inputs. In parallel, feed-forward 3DGS pipelines [2, 4, 39] infer pixel-aligned Gaussians from image features for efficient and high-quality rendering. More recent models [37, 49] advance this direction by directly reconstructing 3D scenes from sparse, unposed images without the need for camera poses, highlighting the potential of learning-based pipelines to generalize in a purely feed-forward manner. Building upon this line of work, we propose a novel 3D stylization pipeline that disentangles structure and appearance, unifying 3D reconstruction and style transfer in a single network.

## 3  Method

Given a set of sparse unposed images $\mathcal{I}^c = \{\mathbf{I}_i^c\}_{i=1}^N$ ($\mathbf{I}_i^c \in \mathbb{R}^{H \times W \times 3}$, where $H$ and $W$ are the height and width of image, superscript $c$ represents content) capturing a scene along with an arbitrary style image $\mathbf{I}^s \in \mathbb{R}^{H \times W \times 3}$ (superscript $s$ represents style), our task is to instantly get the stylized 3D reconstruction of the scene represented by a set of pixel-aligned Gaussians $\mathcal{G}^s = \{\mathbf{g}_j^s\}_{j=1}^{N \times H \times W}$, without compromising multi-view consistency and the underlying scene structure. These Gaussians $\mathcal{G}^s$ are parameterized by $\{(\boldsymbol{\mu}_j, \alpha_j, \mathbf{r}_j, \mathbf{s}_j, \mathbf{c}_j^s)\}_{j=1}^{N \times H \times W}$, where $\boldsymbol{\mu}_j, \alpha_j, \mathbf{r}_j, \mathbf{s}_j$ and $\mathbf{c}_j^s$ are the Gaussian's position, opacity, orientation, scale and stylized color. Alternatively, it is also able to predict the non-stylized Gaussians $\mathcal{G}^c$, which share all the other attributes but have a different set of colors $\{\mathbf{c}_j^c\}_{j=1}^{N \times H \times W}$ compared to $\mathcal{G}^s$.

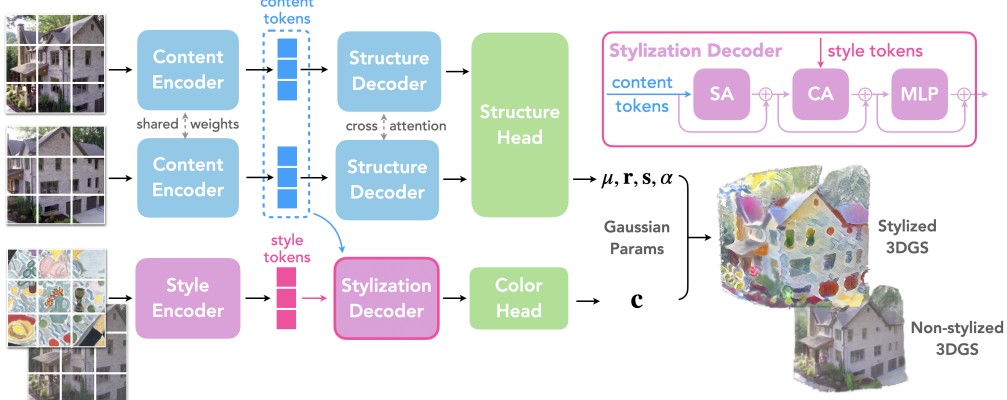

Figure 2: **Overview of Styl3R.** Our model consists of a structure branch and an appearance branch that output different attributes of Gaussians. For the structure branch, sparse unposed images are encoded by a shared content encoder, then content tokens of each image are separately fed into their structure decoders with information sharing between other views. Attributes that govern the structure of the scene are then regressed from structure heads. For the color branch, a style image is encoded by the style encoder, then the output style tokens perform cross attention with content tokens from all viewpoints in the stylization decoder. Finally the color of Gaussians are predicted from these blended tokens output by this decoder, which compose all Gaussian parameters along with the output from structure branch. Apart from style image, the appearance branch can also accept a content image which gives the Gaussians their original colors.

Inspired by [5, 49], we propose a dual-branch architecture that separates the network into a structure building branch and an appearance shading branch. In the appearance branch, we employ a stylization decoder that first performs global self-attention across content tokens from all views to ensure multi-view consistency, then injects style tokens to perform cross-attention with content tokens without interfering the structure branch.

An overview of the pipeline is shown in Fig. 2. In this section, we first introduce the structure branch that leverage dense geometry prior from DUSt3R [45] (Sec. 3.1). Then we illustrate the appearance branch that governs the color of output Gaussians (Sec. 3.2). Finally, we design a training curriculum that facilitate the learning of stylization while effectively preserving geometry prior (Sec. 3.3).

## 3.1 Structure Branch

In order to leverage the dense geometry prior from DUSt3R [45], we employ a ViT-based encoder-decoder architecture to estimate the structure of the scene. A set of sparse unposed images $\mathcal{I}^c$ capturing the scene are first patchified and then encoded by a shared ViT encoder to a set of content tokens $\mathcal{T}^c = \{\mathbf{t}^c_k\}_{k=1}^{N \times \frac{H}{p} \times \frac{W}{p}}$, where $p$ is the patch size. The encoded tokens from one view is then fed into a ViT decoder which perform cross attention with concatenated tokens from all other views to ensure channeling of multi-view information as in [49]. From these output tokens of the decoder for each view, one DPT head [33] is employed to predict the center positions $\boldsymbol{\mu}_j$ of Gaussians, another DPT head regresses other structural attributes: orientations $\mathbf{r}_j$, scales $\mathbf{s}_j$ and opacities $\alpha_j$.

## 3.2 Appearance Branch

To separate the colorization of Gaussians from the estimation of scene structure, we propose an appearance branch that ensures the subsequent stylization would not degrade the learned geometry prior in the structure branch. For ease of alignment between encoded content and style tokens, we leverage the same architecture as content image encoder for style image encoder, but with a different set of learnable weights. This ViT-based style encoder receives an arbitrary style image $\mathbf{I}^s$ and then outputs a set of style tokens $\mathcal{T}^s = \{\mathbf{t}^s_m\}_{m=1}^{\frac{H}{p} \times \frac{W}{p}}$, which are then sent to the stylization decoder.

**Stylization Decoder.** Within the stylization decoder, content tokens $\mathcal{T}^c$ output by the content encoder are stylized by style tokens $\mathcal{T}^s$. Initially, $\mathcal{T}^c$ from multiple views are concatenated, and then

perform a global self-attention to get $\hat{\mathcal{T}}^c$ ensuring multi-view consistency. Then these self-attended content tokens $\hat{\mathcal{T}}^c$ are used to generate queries while style tokens $\mathcal{T}^s$ are used to generate keys and values for the cross attention that blends these two streams of information as shown in Eq. 1.

$$\mathcal{T}^{cs} = \texttt{CrossAttention}(\hat{\mathcal{T}}^c W^Q, \mathcal{T}^s W^K, \mathcal{T}^s W^V) \tag{1}$$

where $W^Q$, $W^K$ and $W^V$ are the projection matrices to generate queries, keys and values for cross attention. After this blending, stylization decoder finally outputs a set of stylized content tokens $\mathcal{T}^{cs}$.

**Color Head.**  From these stylized content tokens $\mathcal{T}^{cs}$, a DPT head is used to predict the stylized color $\mathbf{c}_j^s$ for each Gaussian, which is adapted according to the given style image $\mathbf{I}^s$. These color components $\{\mathbf{c}_j^s\}_{j=1}^{N \times H \times W}$ along with the other parameters regressed from the structure branch compose the complete set of attributes $\{(\boldsymbol{\mu}_j, \alpha_j, \mathbf{r}_j, \mathbf{s}_j, \mathbf{c}_j^s)\}_{j=1}^{N \times H \times W}$ for stylized Gaussians $\mathcal{G}^s$.

**Content as Style.**  Worth noticing, a content image $\mathbf{I}_i^c$ can also be viewed as a special style image that maps the content to its original appearance. This naturally leads the stylization of appearance branch to normal photorealistic shading in novel view synthesis. Thus, inputting a content image $\mathbf{I}_i^c$ to style branch will give us the non-stylized Gaussians $\mathcal{G}^c$. This insight is applied to facilitate subsequent training.

### 3.3  Training Curriculum

Notably, 3D stylization and reconstruction are not inherently well aligned, as optimizing for style loss may degrade the underlying 3D structure of the scene [8]. To address this, we adopt a two-stage training curriculum. In the first stage, the model is trained to accurately estimate the scene structure and perform standard photorealistic shading. After this stage, we proceed to a stylization fine-tuning stage, during which the structure branch is frozen to ensure faithful preservation of the scene geometry.

**Novel View Synthesis Pre-training.**  At this stage, we train the whole model end-to-end for the novel view synthesis (NVS) by solely using photometric loss calculated between novel view images rendered from $\mathcal{G}^c$ and ground truth target RGB images as in Eq. 2. During NVS training, we randomly input one content image $\mathbf{I}_i^c$ to the appearance branch. This encourages the appearance branch to preserve the original scene color at this stage. After this pre-training phase, given a set of sparsely unposed images, the structure branch can predict intricate 3D structure, while the appearance branch is capable of performing photorealistic shading for Gaussians, which lays a solid foundation for the next stylization fine-tuning stage.

**Stylization Fine-tuning.**  Building upon the preceding novel view synthesis (NVS) pre-training, the model can primarily focus on learning to stylize the appearance of the Gaussians. In each forward pass, we input content images $\mathcal{I}^c$ and a style image $\mathbf{I}^s$ to the network, which outputs the stylized Gaussians $\mathcal{G}^s$ to render stylized images at novel viewpoints. These images are used to calculate the losses in Eq. 2 to update the appearance branch. As mentioned in [8], optimizing all Gaussians parameters towards style loss can drastically degrade the structure of reconstructed scene, thus we only fine-tune the appeareance branch during this stage, and freeze the structure branch.

In terms of loss functions, we first employ a weighted combination of style and content losses. For the style loss, we measure the differences in mean and variance between novel view images rendered from $\mathcal{G}^s$ and the reference style image $\mathbf{I}^s$, computed over the `relu1_1`, `relu2_1`, `relu3_1`, and `relu4_1` feature maps of VGG19 [36]. For the content loss, we compare the `relu3_1` and `relu4_1` feature map responses between images rendered from $\mathcal{G}^s$ and the corresponding ground-truth target RGB images. Empirically, we find that incorporating both `relu3_1` and `relu4_1` in the content loss more effectively preserves the structural fidelity of the original scene, compared to using a single layer as commonly done in previous style transfer approaches [8, 54], as shown in Fig. 7.

Besides, to preserve the model's NVS ability during stylization fine-tuning, we adapt the identity loss from [31]. Apart from the style image $\mathbf{I}^s$, we also feed a randomly selected content image $\mathbf{I}_i^c$ to the appearance branch to obtain the non-stylized Gaussians $\mathcal{G}^c$. Similar to the first stage, we also minimize the photometric loss between novel view images rendered from $\mathcal{G}^c$ and ground truth target RGB images while optimizing style and content losses.

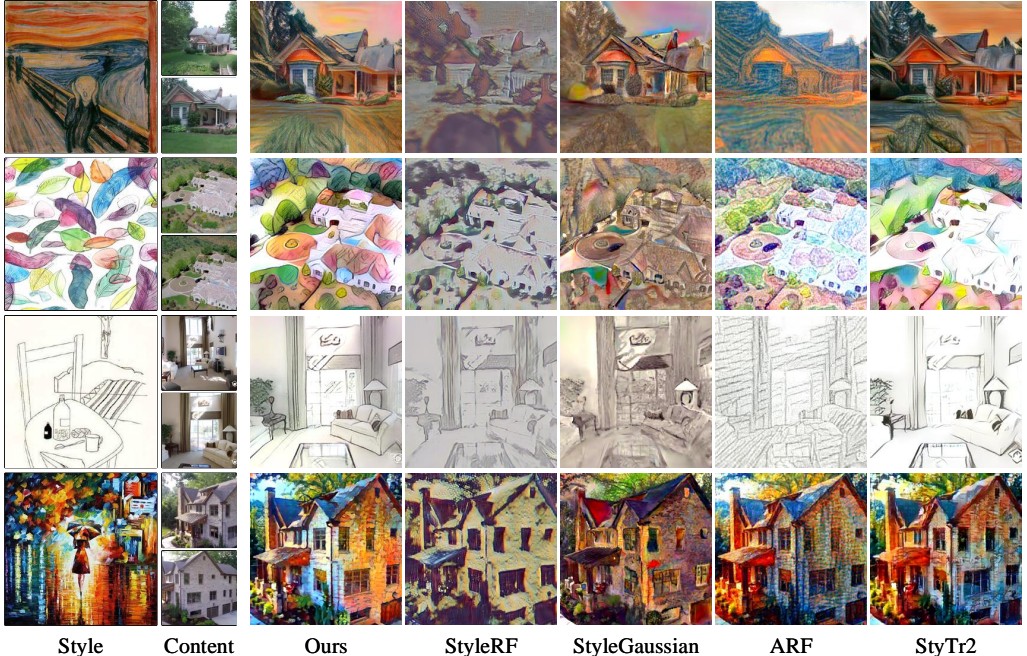

| Style | Content | Ours | StyleRF | StyleGaussian | ARF | StyTr2 |

Figure 3: **Novel View Transfer Comparision on RE10K.** Despite limited image overlap, our method generates stylized novel views that more faithfully capture style details while preserving the original scene structure. In comparison, StyleRF [23] and StyleGaussian [24] tend to produce over-smoothed results that deviate from the true color tone of the reference style. ARF [54] suffers from style overflow, leading to significant loss of content appearance. As a 2D baseline, StyTr2 [5] operates directly on ground-truth novel views, but fails to retain fine structural details of the scene.

**Training Losses.** The losses used in two training stages are summarized as below.

$$\mathcal{L} = \begin{cases} \mathcal{L}_{\text{photo}}(\mathcal{G}^c), & \text{NVS pre-training} \\ \lambda \mathcal{L}_{\text{style}}(\mathcal{G}^s) + \mathcal{L}_{\text{content}}(\mathcal{G}^s) + \mathcal{L}_{\text{photo}}(\mathcal{G}^c), & \text{stylization fine-tuning} \end{cases} \quad (2)$$

where $\mathcal{L}_{\text{photo}}$ is the photometric loss which is a linear combination of MSE and LPIPS [55] loss with weights of 1 and 0.05, respectively, and $\lambda = 10$ is the weight for style loss.

**Progressive Multi-view Training** To stabilize multi-view training, we first pre-train the model on the 2-view setting for the NVS task, which is then used to initialize the 4-view NVS training and subsequent stylization fine-tuning. Though trained with 4 input views, our model can flexibly handle 2 to 8 views during inference as shown in Fig. 8.

## 4  Experiments

**Datasets.** We use a combination of RealEstate10K (RE10K) [56] and DL3DV [22] as our scene dataset, covering both indoor and outdoor videos with diverse camera motion patterns. For style supervision, we use WikiArt [32], and assign a unique style image to each scene in the training and evaluation sets. This setup ensures that neither the test scenes nor styles were seen during training.

To evaluate zero-shot generalization, we test on the Tanks and Temples [16] dataset which is widely used by prior 3D style transfer methods [8, 23, 24, 54].

**Baselines.** Since no existing methods can instantly stylize 3D reconstructions from sparse, unposed content images and a style reference image (as outlined in Table 1), we carefully select a set of representative baselines for comparison. For 2D-based approaches, we adopt a two-stage pipeline using AdaIN [12], AdaAttN [25], and StyTr2 [5]: we first extract ground-truth novel view images and then apply each 2D stylization model to these images. For 3D-based methods, we compare against ARF [54], StyleRF [23], and StyleGaussian [24], which perform 3D stylization but require dense input views and test-time optimization. To ensure proper functionality, we train these methods

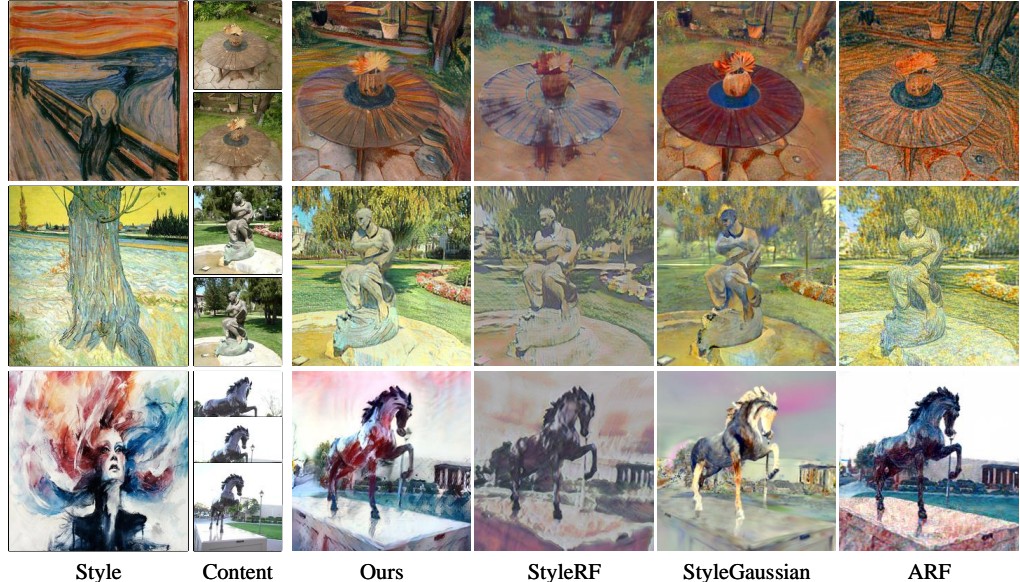

| Style | Content | Ours | StyleRF | StyleGaussian | ARF |

Figure 4: **Cross-dataset generalization on Tanks and Temples dataset.** Our model achieves superior or comparable zero-shot style transfer on out-of-distribution data, outperforming style-free baselines such as StyleRF [23] and StyleGaussian [24] that require per-scene optimization, and matching the performance of ARF [54], which further demands per-scene and per-style optimization.

|  | Method | Consistency | | | | Stylization Time |
|---|---|---|---|---|---|---|
|  |  | Short-range | | Long-range | | |
|  |  | LPIPS↓ | RMSE↓ | LPIPS↓ | RMSE↓ | |
| 2D | AdaIN [12] | 0.163 | 0.063 | 0.323 | 0.111 | 0.004 s |
|  | AdaAttN [25] | 0.224 | 0.071 | 0.331 | 0.098 | 0.024 s |
|  | StyTr2 [5] | 0.167 | 0.059 | 0.315 | 0.098 | 0.029 s |
| 3D | StyleRF [23] | 0.062 | **0.021** | 0.172 | 0.042 | 90 mins |
|  | StyleGS [24] | 0.048 | 0.022 | 0.137 | 0.043 | 132 mins |
|  | ARF [54] | 0.093 | 0.038 | 0.217 | 0.070 | 12 mins |
|  | Styl3R(Ours) | **0.044** | 0.022 | **0.107** | **0.038** | 0.147 s |

Table 2: **Quantitative Results.** Performance comparison of Styl3R with 2D and 3D baselines on RE10K in terms of view consistency. Stylization time refers to processing time excluding IO time.

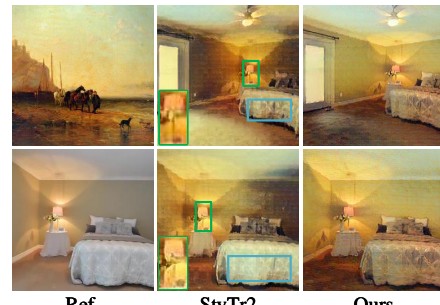

| Ref. | StyTr2 | Ours |

Figure 5: **Visual Comparison.** Visualizations of different views produced by StyTr2 [5] and our method. The highlighted regions (lamp and bed sheet) show noticeable color discrepancies in StyTr2, while our approach maintains consistent color across views.

with dense inputs, acknowledging that this gives them an advantage and makes the comparison less favorable to our approach, which requires only sparse input views. Specifically, ARF [54] requires both per-scene and per-style optimization; while StyleRF [23] and StyleGaussian [24] support zero-shot style transfer, they still depend on per-scene optimization.

**Evaluation Metrics.** Because of the novel and under-explored nature of 3D stylization, there are few metrics for assessing the quality of the stylization. Therefore, we evaluate the multi-view consistency as in prior 3D stylization works [8, 23, 24]. We estimate optical flow between sequential images using RAFT [41], then warp the earlier frame with softmax splatting [29]. Consistency is measured by LPIPS [55] and RMSE between the warped and target images over valid pixels. Short- and long-range consistency are computed between adjacent views and those seven frames apart, respectively. We further employed ArtFID [46], a metric well aligned with human perceptual judgment by jointly assessing content preservation and style fidelity, together with the RGB-uv histogram from HistoGAN [1] to comprehensively evaluate the quality of color transfer. To evaluate novel view synthesis quality, we report standard image similarity metrics: PSNR, SSIM, and LPIPS [55].

| Method | PSNR ↑ | SSIM ↑ | LPIPS ↓ |
|---|---|---|---|
| pixelSplat [2] | 23.848 | 0.806 | 0.185 |
| MVSplat [4] | 23.977 | 0.811 | 0.176 |
| NoPoSplat [49] | 25.033 | 0.838 | 0.160 |
| NoPoSplat* | 24.836 | 0.832 | 0.166 |
| Ours | 24.871 | 0.837 | 0.165 |
| Ours-stylization | 24.055 | 0.820 | 0.179 |

Table 6: **NVS comparison on RE10K.** * denotes 0-degree spherical harmonics, as used in our model, while [49] defaults to 4-degree.

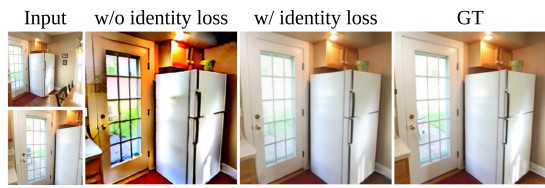

Figure 6: **Ablations.** NVS results w/o and w/ identity loss during stylization fine-tuning. The former fails to retain the true color tone of the scene.

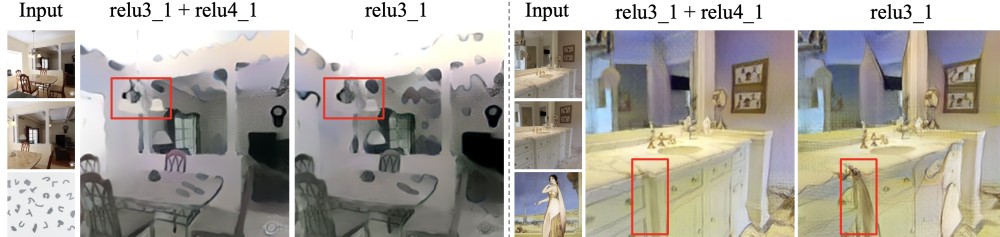

Figure 7: **Ablations.** Stylization results of model trained with content loss consist of different layers. Using `relu3_1` and `relu4_1` in content loss preserve the original scene appearance more faithfully.

**User Study.** We conducted an online user study to evaluate perceptual preferences between our proposed approach and alternative methods. Participants were asked to select the result they perceived as exhibiting the most harmonious integration of style and scene. In total, approximately 2,000 votes were collected, including 1,274 for RE10K, 294 for Tanks and Temples, and 392 for the ablation study.

**Implementation details.** We use Py-Torch. The content and style encoder adopts a standard ViT-Large architecture with a patch size 16, while the structure and stylization decoder is based on a ViT-Base model. We initialize the encoder, decoder, and the Gaussian center prediction head with pretrained weights from MASt3R [19], whereas the remaining layers are initialized randomly. The model is trained on images with a resolution of $256 \times 256$. Besides, we use 0 degree spherical harmonics for Gaussians following [8]. Training takes ~1.5 days on 8 NVIDIA A100 GPUs.

|  | Styl3R | ARF | StyleGS | StyleRF | StyTr2 |
|---|---|---|---|---|---|
| RE10K (%) | **53.29** | 14.13 | 6.67 | 1.81 | 24.10 |
| TnT (%) | **38.78** | 22.11 | 11.56 | 3.06 | 24.49 |

Table 3: **User Study.** Voting results for stylization.

| Metric | Styl3R | ARF | StyleGS | StyleRF | StyTr2 |
|---|---|---|---|---|---|
| ArtFID↓RE10K | **35.12** | 42.95 | 55.75 | 46.59 | 38.93 |
| ArtFID↓TnT | **38.05** | 48.98 | 55.03 | 64.81 | 39.24 |
| H-gram↓RE10K | **0.230** | 0.313 | 0.465 | 0.507 | 0.241 |
| H-gram↓TnT | **0.241** | 0.259 | 0.379 | 0.422 | 0.247 |

Table 4: **ArtFID [46] and Histogram [1] for comparisons.**

|  | ArtFID ↓ | Histogram ↓ | Votes (%) |
|---|---|---|---|
| h3 layer | 42.01 | 0.277 | 35.20 |
| h3+h4 layer | **35.83** | **0.239** | **64.80** |

Table 5: **Ablations.** The ArtFID, Histogram, and voting results for h3 and h3+h4 layer.

## 4.1 Experimental Results

**3D Stylization Results.** As shown in Fig. 3, Table 2, Table 3 and Table 4, our method outperforms all baselines qualitatively and quantitatively. Visually, our stylizations achieve a more balanced trade-off between content preservation and faithful style transfer. Among test-time optimization-based 3D baselines, StyleRF [23] and StyleGaussian [24] often fail to reproduce the reference style's color tone accurately, resulting in overly whitened or darkened outputs. ARF [54], while better at capturing style colors, tends to overfit and apply excessive stylization that obscures scene details. For example, in the third row of Fig. 3, furniture in the living room becomes nearly indistinguishable due to heavy sketch-line artifacts. As a 2D baseline, StyTr2 [5] generates visually pleasing results on individual ground-truth novel views, but it lacks multi-view consistency, as evidenced in Table 2 and Fig. 5. In contrast, our method consistently produces superior stylizations while maintaining the best short- and long-range consistency metrics, benefiting from our attention mechanism that operates jointly over multi-view content and style tokens. Although StyleRF achieves a slightly lower RMSE in short-range evaluations, this is largely due to its over-smoothed outputs, as clearly illustrated in Fig. 3.

| Input | 2-view model | 4-view model | Input | 2-view model | 4-view model |

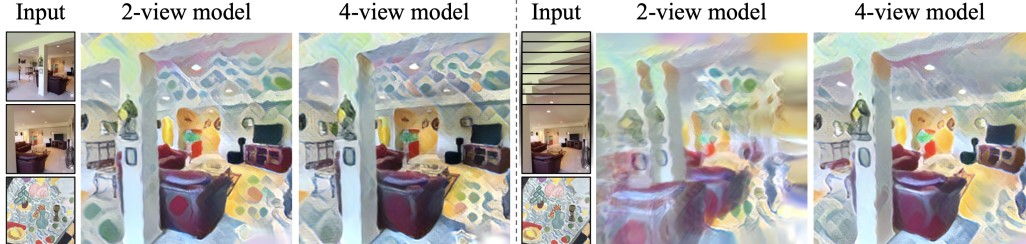

Figure 8: **Ablations.** Results for 2-view and 4-view models when inputting 2 and 8 content images. The 2-view model fails to align Gaussians across multiple views when provided with 8 input views.

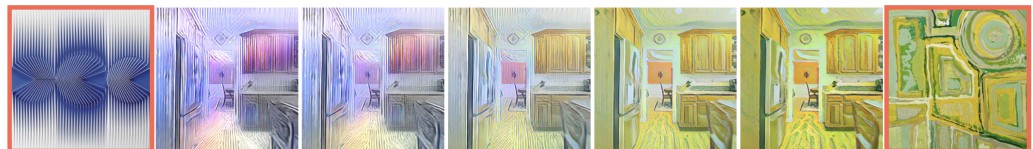

Figure 9: **Application.** Style Interpolation with 2 style images by interpolating their style tokens. It can be observed that the style of the scene smoothly transit from one to another.

**Cross-Data Generalization.** To evaluate the generalization performance of our method, we directly apply it to the Tanks and Temples dataset [16], a widely used benchmark in prior works. As shown in Fig. 4, our model demonstrates superior performance on out-of-distribution scenes such as *Garden*, *Ignatius*, and *Horse* which are object-centric scenes that differ significantly from the RE10K training data, outperforming existing state-of-the-art methods. Notably, even though StyleRF [23] and StyleGaussian [24] are trained per scene, they fail to generalize to arbitrary style inputs. While ARF [54] achieves better results in some scenes, it requires dense, calibrated views along with per-scene and per-style optimization, limiting its practicality for time-constrained applications.

**Novel View Synthesis.** Our final model supports both stylized and standard 3D reconstruction, depending on whether the input to the appearance branch is a style or content image. We report two sets of metrics: one for the stylized output (Ours-stylization) and one for standard reconstruction without stylization fine-tuning (Ours). As shown in Table 6, Ours achieves performance comparable to NoPoSplat [49], despite not initializing the stylization decoder with pretrained weights. While Ours-stylization shows a slight drop in performance, it enables simultaneous support for both photorealistic and stylized reconstruction. Our results are from 2-view RE10K models, consistent with NoPoSplat.

**Stylization Time.** We define stylization time as the total time from receiving the input content and style images to producing the final stylized outputs. This metric more practically reflects how quick a user can obtain stylized results. For 3D methods, this includes both the reconstruction time and any stylization-related training or optimization. As shown in Table 2, our method achieves significantly faster stylization time than all existing 3D approaches, while approaching state-of-the-art 2D methods in terms of speed.

## 4.2 Ablation Studies

**Identity Loss to Preserve NVS Capability.** We explore the necessity of identity loss during stylization fine-tuning. It can be observed in Fig. 6 that the model fails to recover the original appearance of the scene while performing NVS if we disable this loss.

**Content Loss Layers.** As in Sec. 3.3, using both `relu3_1` and `relu4_1` for the content loss better preserves structural details without sacrificing artistic expression. As shown in Fig. 7 and Table 5, relying solely on `relu3_1` tends to cause the style to overwhelm the underlying scene structure.

**Flexibility of Number of Input Views.** As discussed in Sec. 3.3, our model trained with 4 content images demonstrates strong generalization, effectively handling between 2 to 8 input views. As in Fig. 8, both the 2-view and 4-view models produce satisfactory stylizations when given only 2 content images. However, when the input is increased to 8 content images, the 2-view model struggles to align Gaussians across views, resulting in duplicated artifacts such as multiple pillars and sofas. In contrast, the 4-view model performs remarkably well, despite never being trained with 8-view inputs.

### 4.3 Application

**Style Interpolation.** We demonstrate an application of our model, style interpolation, in Fig. 9. Specifically, we interpolate the style tokens from two reference style images before passing them to the stylization decoder, producing a blended stylization that smoothly transitions between the two styles. This approach can be easily extended to more than two styles by simply computing a weighted sum of their respective style tokens.

## 5 Discussion

To clarify the motivation of our method, we provide a detailed comparison between two-stage approaches, which first reconstruct 3DGS and then perform stylization, and our single-pass method that achieves 3D stylization in an end-to-end manner.[2]

**Two-stage methods:**

- **(+) High-fidelity geometry:** The 3DGS model is optimized directly to fit the target scene from dense image inputs, resulting in accurate geometry and detailed structures.
- **(+) Scalable to large scenes:** With sufficient optimization time and resources, this approach can be extended to larger-scale environments.
- **(–) No instant stylization support:** There is currently no method that directly takes a trained 3DGS and outputs a stylized version without additional test-time optimization. Even the closest work (e.g., StyleGaussian) requires post-processing like VGG feature embedding.
- **(–) Inefficient:** The optimization-heavy pipeline is unsuitable for interactive or time-sensitive applications such as AR or mobile capture.
- **(–) Requires expert setup:** Users must collect dense, accurately posed input images to fit 3DGS properly, limiting accessibility for casual creators.

**Single-pass methods (ours):**

- **(+) Fast inference:** The model runs in a single forward pass with no test-time optimization, enabling nearly real-time 3D stylization—ideal for interactive or mobile scenarios.
- **(+) User-friendly:** The pipeline accepts a small set of casually captured images and produces a stylized 3D scene in an end-to-end manner, making it accessible to non-experts.
- **(–) Data-hungry:** Training requires a large and diverse dataset to generalize across various scene types and artistic styles.
- **(–) Lower reconstruction quality:** Feed-forward models typically prioritize speed and generalization over precise geometric fidelity.
- **(–) Limited scalability:** Current implementations may face memory constraints (e.g., VRAM bottlenecks) and require architectural adjustments or more training data to handle large-scale environments.

## 6 Conclusion

This paper introduces a feed-forward network for instant 3D stylization from sparse, unposed input views and a single reference style image, which generalizes to arbitrary scenes and styles without test-time optimization. The network is composed of a structure branch and an appearance branch, jointly enabling consistent novel view synthesis and stylization. Extensive experiments demonstrate that our method outperforms existing baselines in zero-shot stylization quality, while achieving significantly faster inference speed, making it more practical for real-world and interactive applications. It is worth noting that our method currently supports only static scenes; extending it to handle dynamic scenarios is an important direction for future work.

**Acknowledgements.** This work was supported in part by NSFC under Grant 62202389, in part by a grant from the Westlake University-Muyuan Joint Research Institute, and in part by the Westlake Education Foundation.

---

[2]Throughout this section, "(+)" and "(–)" denote advantages and disadvantages, respectively.

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

# Appendix

In the supplementary, we provide the following:

- more implementation details, including architecture and training hyperparameters for our model in Sec. A;

- more technical details on training 2D and 3D baselines in Sec. A;

- more visual results of our model and comparisons with baselines in Sec. B;

- limitations of our model in Sec. C.

*We highly recommend visiting our* **project website** *for a more comprehensive demonstration of our method's stylization quality and temporal consistency. The website features:*

- qualitative comparisons with baselines on the RE10K [56] and Tanks and Temples [16] datasets;

- more out-of-domain stylization results on the Tanks and Temples and NeRF LLFF [26] datasets;

- style interpolation examples showcasing transitions between three different styles.

## A   More Implementation Details

**Training.**   In terms of optimization, we employ AdamW optimizer. For Novel View Synthesis (NVS) pretraining, we train the stylization decoder, color head and structure head with initial learning rate of $2 \times 10^{-4}$, and fine-tune the other parameters with $2 \times 10^{-5}$. Then for stylization fine-tuning, we continue optimizing the color head and stylization decoder with initial learning rate of $2 \times 10^{-4}$ and fine-tune only the style encoder with $2 \times 10^{-5}$, and keep all the other parameters in the structure branch fixed.

**Architecture.**   To expedite the inference of network, we use the flash attention implementation from `xFormers` [18] in all of our encoders and decoders. As in [49], we feed the tokens from the 1-st, 7-th, 10-th and 13-rd block into DPT [33] for upsampling.

**Baselines Training.**   For the 2D methods (StyTr2 [5], AdaIN [12], AdaAttN [25]), we directly utilize their publicly released pretrained checkpoints, available at StyTr2 code , AdaIN code, and AdaAttN code, respectively. As these methods do not support 3D reconstruction from 2D images, we apply them directly to stylize the ground-truth 2D novel views, bypassing any reconstruction process.

In contrast, the 3D methods (ARF [54], StyleRF [23], StyleGaussian [24]) are unable to reconstruct geometry from sparse, unposed inputs. Therefore, we train them using all available scene images (on average, more than 100 per scene) along with their corresponding camera poses. This setup results in an unfair comparison with our method, which operates on sparse and unposed inputs.

## B   More Visual Results

We present additional qualitative results in Fig. 10, Fig. 11, Fig. 12, Fig. 13, and Fig. 14, which highlight the superior performance of our method compared to prior state-of-the-art style transfer approaches. While existing 3D methods rely on densely posed images, our approach enables instant 3D stylization across arbitrary scenes and styles without such constraints.

**More Comparisons with Baselines.**   To better showcase the superiority of our method, we visualize more comparison results with 3D baseline methods on different scenes and styles, as shown in Fig. 10.

**More Visual Results of Our Method.**   To validate our method is compatible with arbitrary scenes and styles, we show stylization results with exhaustive combinations from randomly selected scenes and styles, as shown in Fig. 11,  Fig. 12,  Fig. 13 and Fig. 14.

## C  Limitations

Inherited from the base model [49], our method is currently limited to 2–8 input views and low-resolution scenarios ($256 \times 256$). The former limitation could be alleviated by replacing the base model with VGGT [43], while the latter could be partially mitigated by fine-tuning the current model on higher-resolution data. However, ultra-high-resolution scenarios remain challenging due to VRAM constraints.

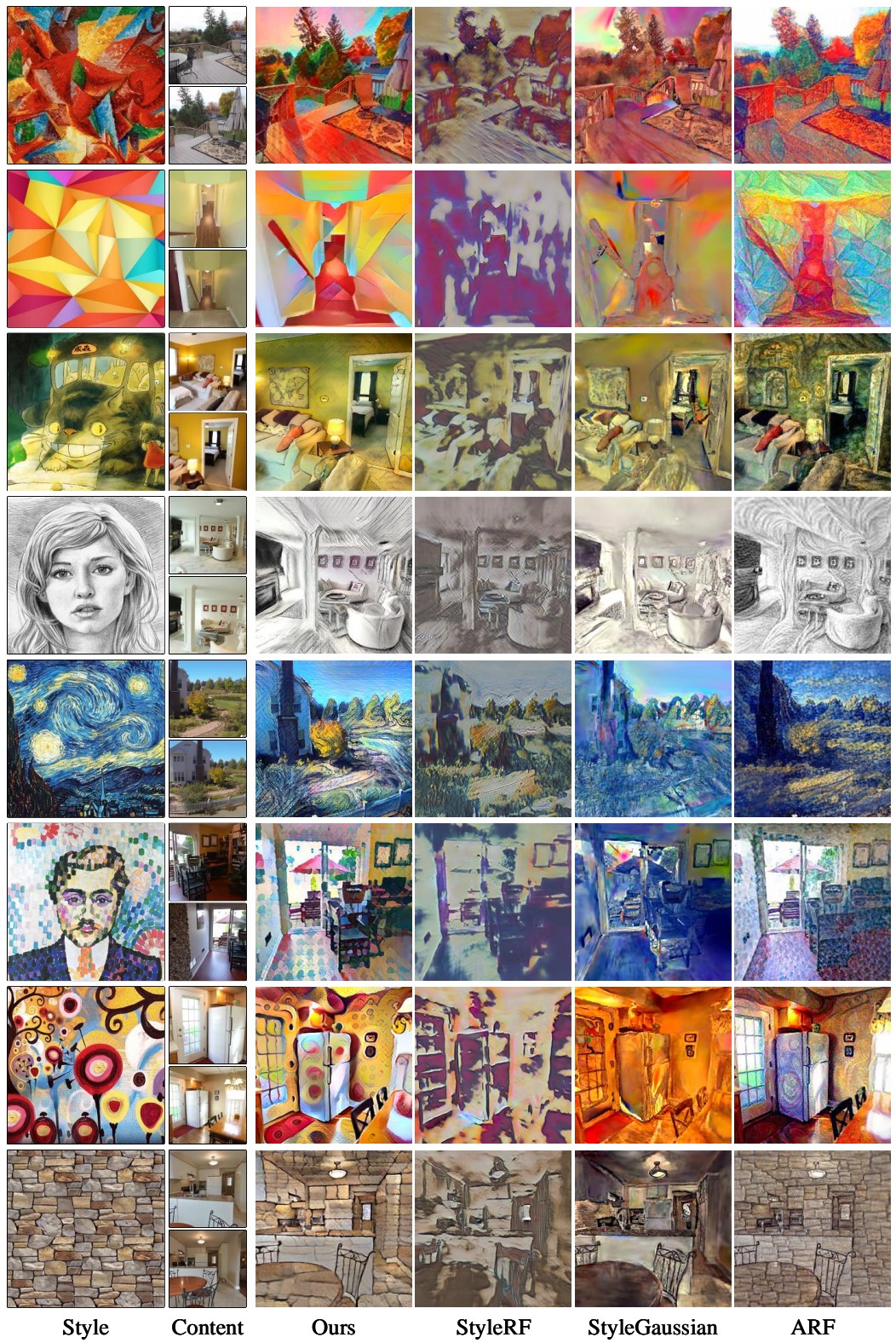

| Style | Content | Ours | StyleRF | StyleGaussian | ARF |

Figure 10: **Novel View Transfer Comparision on RE10K.** Our method faithfully preserves style and scene structure, even with limited image overlap. In contrast, StyleRF [23] and StyleGaussian [24] produce over-smoothed results with inaccurate color tones, while ARF [54] suffers from style overflow.

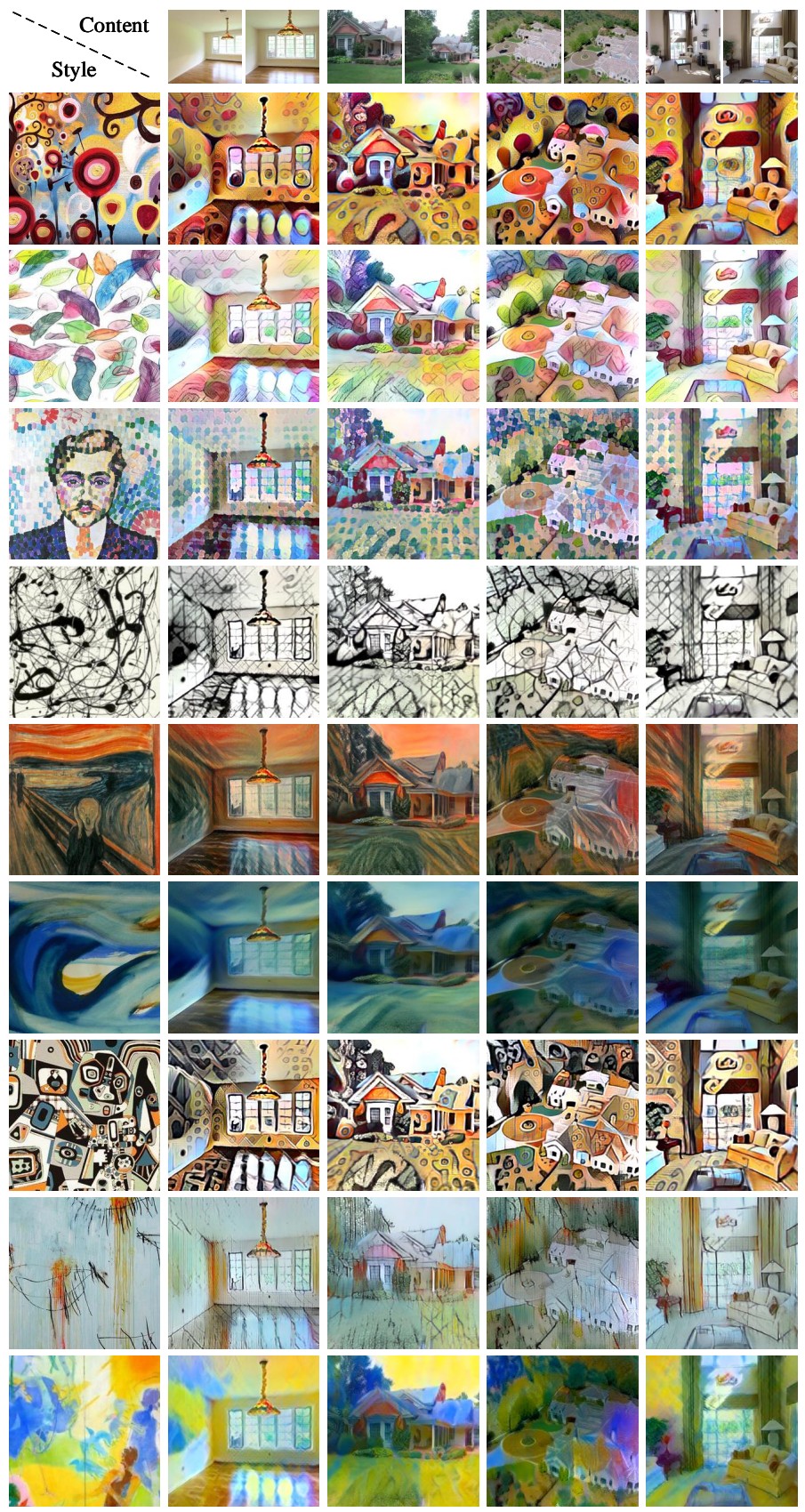

Figure 11: **Additional Results on RE10K from Our Method.**

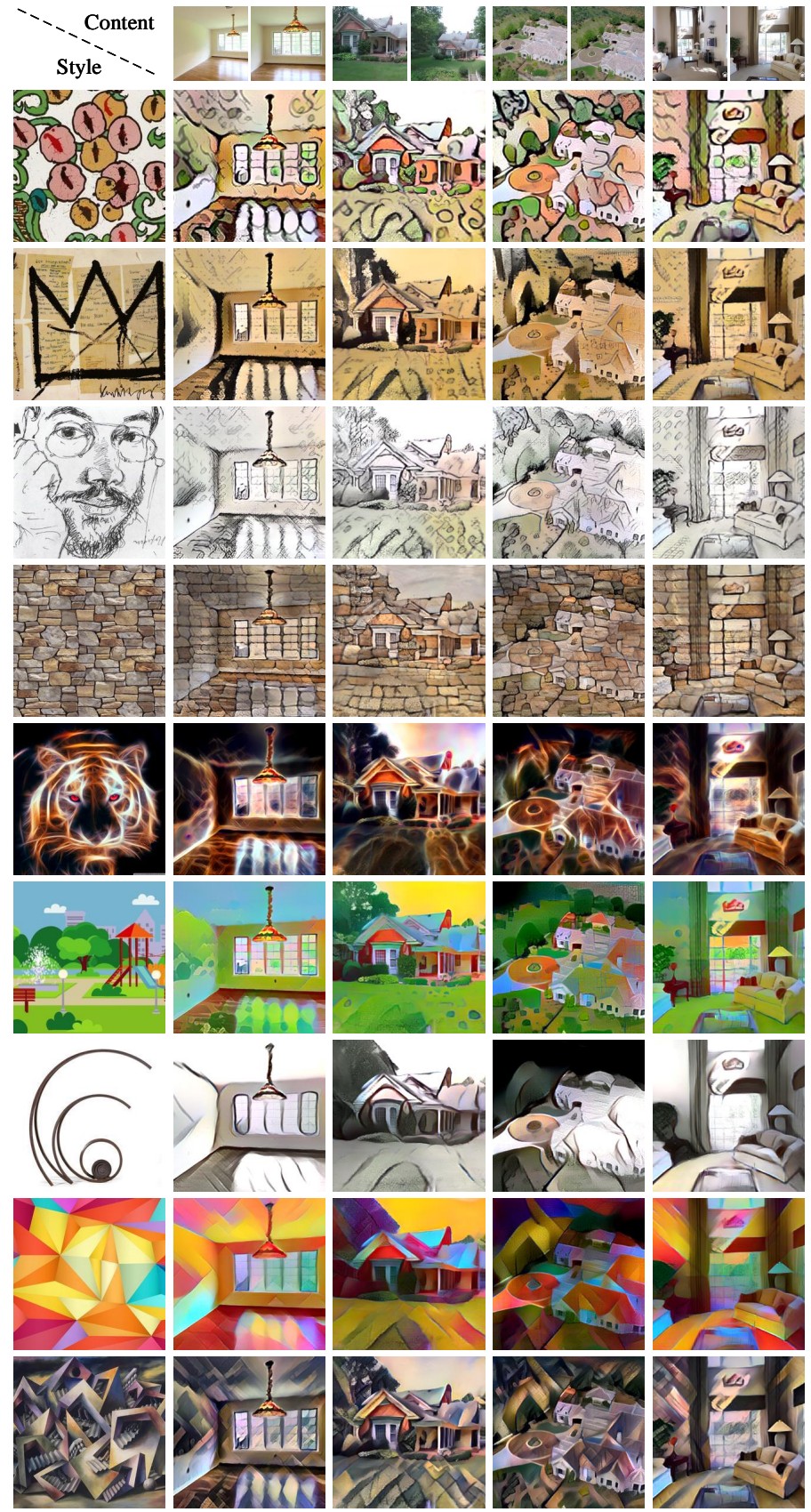

Figure 12: **Additional Results on RE10K from Our Method.**

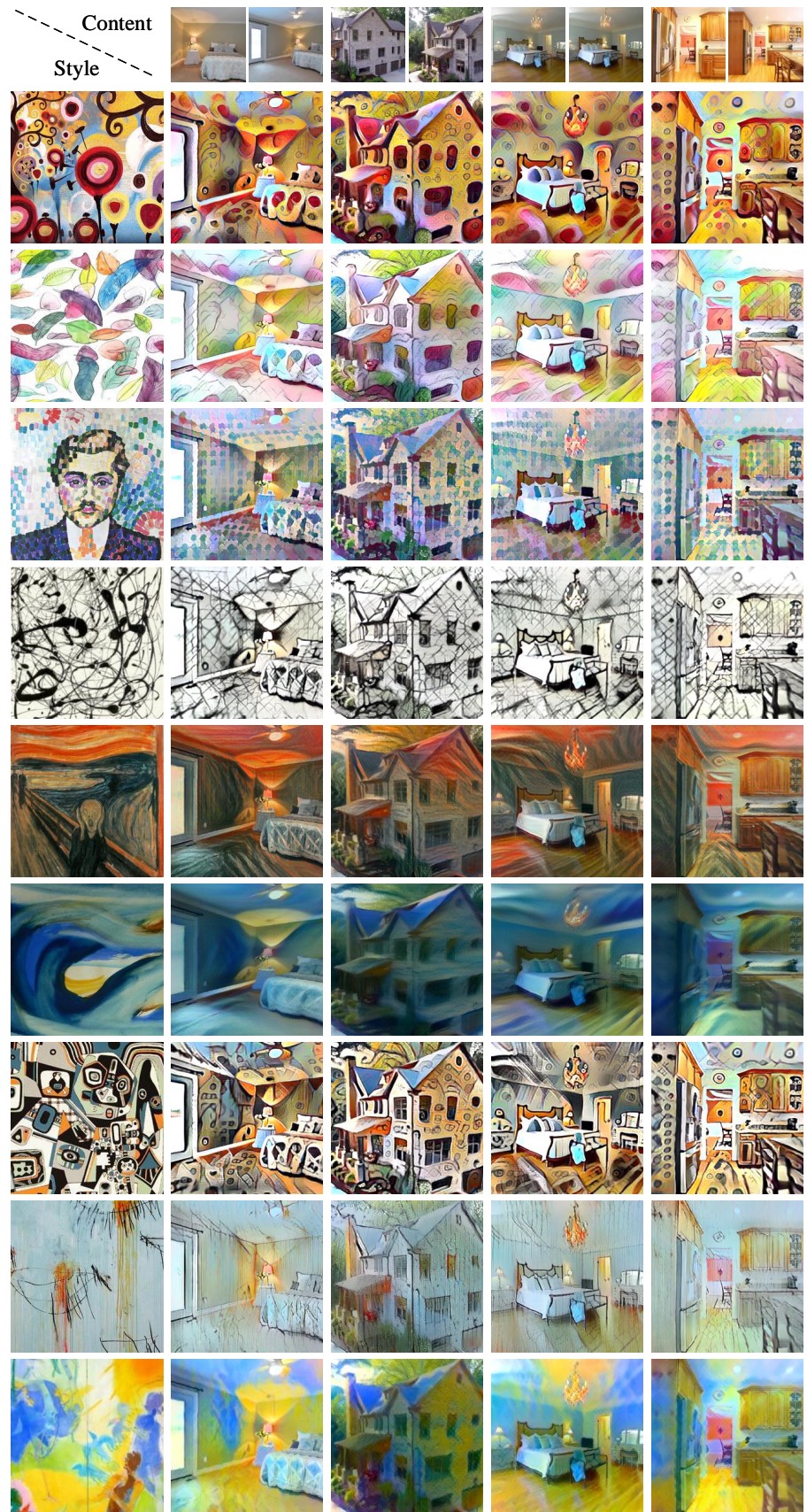

Figure 13: **Additional Results on RE10K from Our Method.**

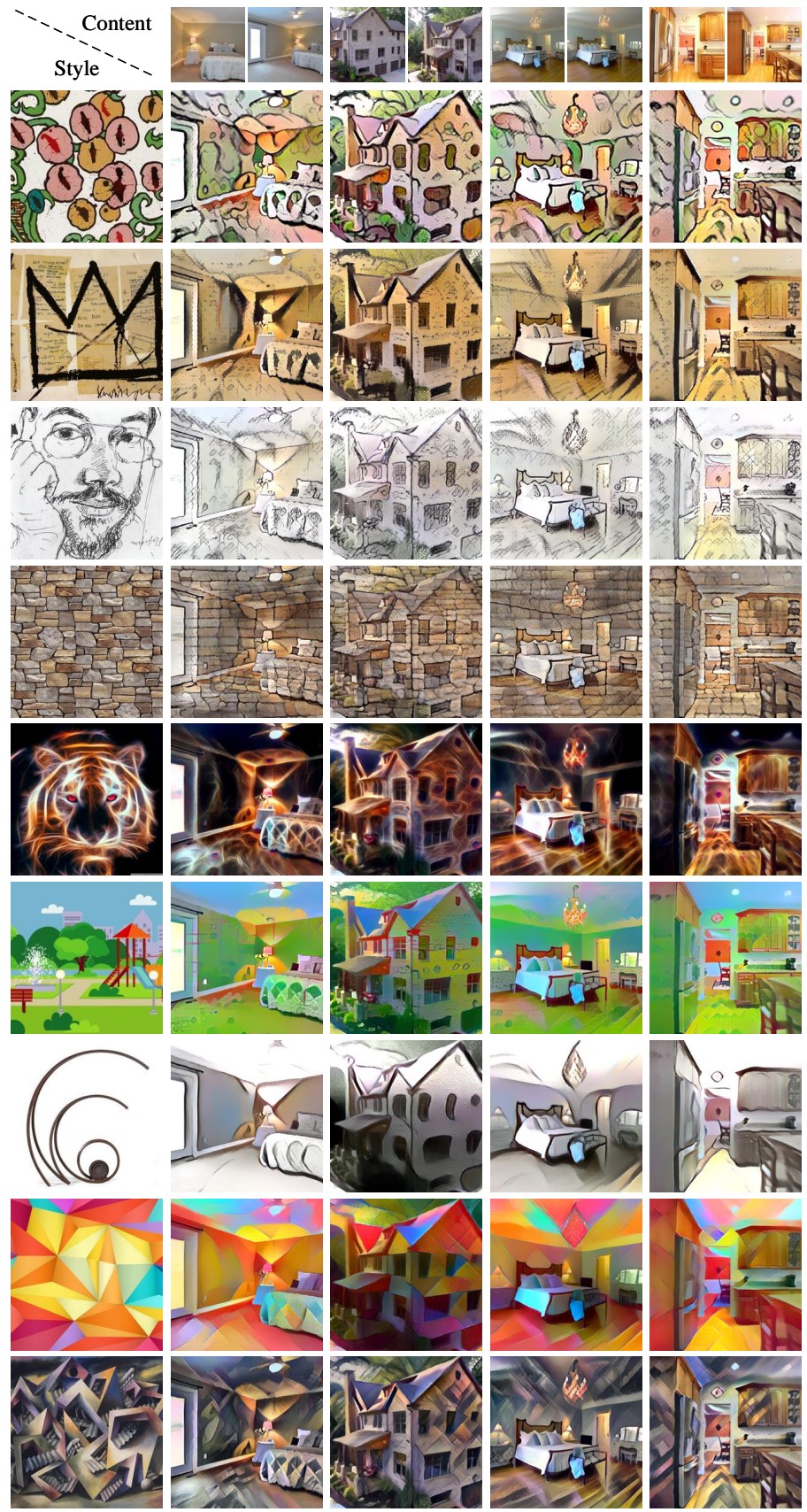

Figure 14: **Additional Results on RE10K from Our Method.**

