# OpenReview forum: "Styl3R: Instant 3D Stylized Reconstruction for Arbitrary Scenes and Styles"
_NeurIPS.cc/2025/Conference — NeurIPS 2025 poster_

### Official Review · Reviewer_4c71 · 2025-06-28

**Clarity:** 3
**Significance:** 3
**Originality:** 3
**Rating:** 5
**Confidence:** 5

**Summary:**

The paper presents ​Styl3R, a feed-forward network for ​instant 3D stylization​ that operates on ​sparse, unposed input views​ (2–8 images) and an arbitrary style image. ​Styl3R utilizes a pipeline similar to NoPoSplat, reconstructing and stylizing 3D scenes in ​under 0.15 seconds​ without test-time optimization or per-scene fine-tuning. The method employs a ​dual-branch architecture​ to decouple structure (3D Gaussian parameters) and appearance (stylized colors), ensuring multi-view consistency while preserving scene geometry. A transformer-based stylization decoder blends style features with multi-view content tokens, and training involves a two-stage curriculum: pre-training for novel view synthesis followed by stylization fine-tuning with an identity loss to retain photorealistic capabilities. Evaluations on multiple datasets demonstrate superior efficiency and quality over prior 3D methods (e.g., StyleRF, ARF) and 2D baselines.

**Questions:**

1. Can the method adjust the stylization strength during inference (e.g., interpolating between original and fully stylized content)?
2. Could the method handle HD/4K resolution inputs, and if so, what architectural modifications would be required?

**Ethical Concerns:**

["NO or VERY MINOR ethics concerns only"]

**Final Justification:**

I appreciate their additional experiments on different resolutions, which demonstrate Styl3R's ability to handle higher-resolution inputs. While Styl3R does have limitations—such as being constrained to a limited number of input views and small-scale scenes—I believe the paper's contributions are strong enough to advance the field of 3D stylization. Styl3R's ability to generalize across scenes and styles while performing stylization at high speed is a significant contribution.

**Limitations:**

yes

**Quality:**

3

**Strengths And Weaknesses:**

Strengths:

1. ​Speed & Practicality: Achieves stylization in ​0.15s, making it suitable for interactive applications. Eliminates the need for dense inputs, camera poses, or per-scene/style optimization.
2. ​Novel Architecture: This is the first work that extends feed-forward 3D reconstruction architecture, such as Dust3R, NoPoSplat, to 3D stylization. ​Transformer-based stylization decoder​ ensures multi-view consistency by blending global style and content features.
3. ​Strong Empirical Results: Outperforms 3D baselines in ​zero-shot stylization​ and matches optimization-heavy methods (e.g., ARF) without their computational overhead. Preserves ​NVS capabilities​ even after stylization fine-tuning (via identity loss).


Weaknesses:​​

1. ​Resolution and aspect ratio limitations:​​ The method is constrained to a fixed 256×256 resolution for both input views and stylized novel views, which restricts its applicability to higher-resolution scenarios.
2. ​Scene scale and camera movement constraints:​​ Based on the presented results, the method appears limited to small-scale scenes and demonstrates only modest camera movements in the rendered stylized views. Unlike approaches such as StyleGaussian, it cannot support free viewpoint exploration of larger environments.

---

> ### Author Rebuttal · Authors · 2025-07-30
>
> We sincerely thank the reviewer for recognizing the *practicality of our work in interactive applications*, the *novelty of our architecture*, and the *quality of our results*. We take all comments seriously and have addressed the reviewer’s concerns in detail below. If there are any remaining questions, we would be happy to clarify further.
>
> > **W1: The method is constrained to a fixed 256×256 resolution for both input views and stylized novel views, which restricts its applicability to higher-resolution scenarios.**
>
> This is not an inherent limitation of our method, but rather a design choice made for faster prototyping. To accelerate development, we trained and tested our model on a lower resolution (256×256). However, our framework is resolution-agnostic and can be scaled up by simply feeding in higher-resolution images.
>
> To evaluate this, we tested the model (trained on 256×256) with input resolutions of 256×352, 352×352, and 352×640. The model produced reasonable results on the first two resolutions, which are closer to the training resolution, and exhibited only slight degradation at 352×640.
>
> We then fine-tuned the model on 352×640 images, which are not only higher in resolution but also differ in aspect ratio. After fine-tuning, the model produced high-quality stylization results, comparable to those results achieved at the lower resolution. This demonstrates the scalability of our method and its potential to handle even higher-resolution scenarios. We would include the results into the final version.
>
> ---
>
> > **W2: Based on the presented results, the method appears limited to small-scale scenes and demonstrates only modest camera movements in the rendered stylized views. Unlike approaches such as StyleGaussian, it cannot support free viewpoint exploration of larger environments.**
>
> It indeeds a limitation (limited number of inputs) of our current model, compared to StyleGaussian. However, Styl3R aims for different scenarios, especially for time-constrained applications (e.g. quick 3D content stylization preview), where StyleGaussian struggles to get the stylized scene quickly. Furthermore, benefit from the fast development of feed-forward 3D reconstruction methods (e.g. a concurrent work from this years CVPR, VGGT could handle more than 100 images), we would foresee Styl3R would inspire follow-up works to handle large-scale scenes in an efficient feed-forward manner in future.
>
> ---
>
> > **Q1: Can the method adjust the stylization strength during inference (e.g., interpolating between original and fully stylized content)?**
>
> Yes, our method allows adjustment of stylization strength at inference time. Similar to style interpolation, we can fix the output of the structure branch and control the stylization level of the appearance by interpolating between the style tokens encoded from the original scene image and the style image.
>
> In a preliminary test, we performed linear interpolation of the style tokens using equal weights (0.5) between the original scene and the style image. The resulting reconstruction exhibits an appearance that lies between the original and fully stylized scenes, demonstrating the model’s ability to support controllable stylization.
>
> ---
>
> > **Q2: Could the method handle HD/4K resolution inputs, and if so, what architectural modifications would be required?**
>
> No, our current model cannot handle HD or 4K resolution inputs without modifying its configuration. Due to the use of a patch size of 16×16, we are limited to a batch size of 2 at a resolution of 352×640 on an A100 GPU with 80GB of VRAM. Simply feeding HD-resolution images would exceed the memory capacity of commonly available GPUs.
>
> To scale up to HD or 4K resolutions, one straightforward approach would be to increase the patch size, thereby reducing the number of tokens extracted from the input images and lowering memory consumption. A more advanced solution would involve applying a chunking strategy [4][5], where the high-resolution image is divided into smaller overlapping chunks, processed independently, and then the resulting Gaussians from each chunk are aligned during post-inference.
>
> [4] Galerne, Bruno, et al. "Scaling Painting Style Transfer." Computer Graphics Forum. Vol. 43. No. 4. 2024.
> [5] Galerne, Bruno, et al. "SGSST: Scaling Gaussian Splatting Style Transfer." CVPR. 2025.

---

> > ### Comment · Reviewer_4c71 · 2025-08-05
> >
> > I thank the authors for their detailed response. I appreciate their additional experiments on different resolutions, which demonstrate Styl3R's ability to handle higher-resolution inputs. While Styl3R does have limitations—such as being constrained to a limited number of input views and small-scale scenes—I believe the paper's contributions are strong enough to advance the field of 3D stylization. Thus I maintain my score as accept.

---

> > > ### Author Response · Authors · 2025-08-05
> > >
> > > Thank you very much for your recognition of our work, and the constructive suggestions on additional evaluations. We would incorporate those additional evaluations and clarifications into the final version of our paper.

---

### Official Review · Reviewer_EHSJ · 2025-06-28

**Clarity:** 3
**Significance:** 3
**Originality:** 3
**Rating:** 4
**Confidence:** 4

**Summary:**

This paper addresses the problem of 3D scene stylization. The authors aim to develop a method that avoids test-time optimization and requires only sparse input images without pose annotations. Building on recent feed-forward reconstruction models, they propose a two-branch architecture that separately models structure and appearance. An identity loss is introduced during pretraining to help the model preserve scene structure when training for stylization. The proposed method achieves high stylization quality and strong multi-view consistency, outperforming prior work on both in-domain and out-of-domain datasets.

**Questions:**

Why are two DPT heads used—one for predicting the center position and another for other attributes? What is the motivation behind this design choice?

**Ethical Concerns:**

["NO or VERY MINOR ethics concerns only"]

**Final Justification:**

Thank you to the authors for the detailed response. My questions have been well addressed, specifically: 1) View coverage: the authors demonstrate that their method can generate results covering 180 degrees of a scene; 2) Stylization quality: the qualitative metrics show clear improvements. In my view, the paper leverages existing techniques to tackle a novel task, which constitutes a meaningful form of innovation. I believe this work could be a valuable contribution to the field.

**Limitations:**

Yes

**Quality:**

3

**Strengths And Weaknesses:**

Strengths:
1. The paper addresses a novel problem of 3D scene style transfer using sparse input without known camera poses, which, to the best of my knowledge, has not been explored before.

2. While using feedforward networks for fast 3D reconstruction has recently gained attention, this paper presents the first framework that applies feedforward networks to 3D scene stylization. The approach is both novel and interesting. The proposed framework is clean and straightforward.

3. Compared to prior works on 3D scene stylization, the proposed method achieves higher stylization quality, better consistency, and improved efficiency under sparse-view settings.

Weaknesses and Concerns:
1. The paper demonstrates that the model can be tested with 2–8 views. Does training require a fixed number of input images, or is the model trained with variable input sizes? How does the model generalize across different numbers of views? Was this also explored in NoPoSplat?

2. Since the method operates with only 2–8 input images, the viewpoint diversity in the results is quite limited. The paper could be strengthened by showing that increasing the number of input views leads to more diverse outputs, possibly even covering 360-degree viewpoints.

3. In the qualitative results in Figure 3, the second and third rows contain some dark regions, and compared to the 2D style transfer output from StyTr2, the 3D method appears to produce slightly lower stylization quality. Could the authors explain why the proposed method may fall short of matching 2D stylization quality in such cases?

---

> ### Author Rebuttal · Authors · 2025-07-30
>
> We sincerely thank the reviewer for recognizing the *novelty of our problem setting*, as well as the *simplicity and effectiveness of our proposed framework*. We take all comments seriously and have addressed the reviewer’s concerns in detail below. Please let us know if any questions remain — we would be happy to provide further clarification.
>
> > **W1: The paper demonstrates that the model can be tested with 2–8 views. Does training require a fixed number of input images, or is the model trained with variable input sizes? How does the model generalize across different numbers of views? Was this also explored in NoPoSplat?**
>
> Our model is trained with 4 input views and demonstrates flexible generalization to a range of 2 to 8 views. While NoPoSplat briefly mentions generalization with 3 input views in its appendix, it does not thoroughly explore scenarios with more views in their paper.
>
> ---
>
> > **W2: Since the method operates with only 2–8 input images, the viewpoint diversity in the results is quite limited. The paper could be strengthened by showing that increasing the number of input views leads to more diverse outputs, possibly even covering 360-degree viewpoints.**
>
> As suggested by the reviewer, we have tested our pretrained model with 12 input images, the model can effectively cover approximately 180 degrees of viewpoint for some scenes, but is currently unable to handle full 360-degree coverage. This limitation could potentially be mitigated by training the model with an even larger dataset containing richer camera viewpoints. However, due to constraints on the available datasets and time limit during this rebuttal period, we were unable to pursue this direction and would try to include in the final version.
>
> ---
>
> > **W3: In the qualitative results in Figure 3, the second and third rows contain some dark regions, and compared to the 2D style transfer output from StyTr2, the 3D method appears to produce slightly lower stylization quality. Could the authors explain why the proposed method may fall short of matching 2D stylization quality in such cases?**
>
> Thank you for pointing this out—indeed, the 3D and 2D methods exhibit different characteristics in their stylization results.
> In the second and third rows, the 3D method (Styl3R) tends to better preserve the original appearance of the scene, including shading and structural details. It could be caused by the the usage of content image as style image (i.e. identity loss)  during training, to enable the model to also have the normal NVS capability. In contrast, the 2D method (StyTr2) does not have such restrictions, and leans more heavily toward the style image’s appearance, which can result in the loss of scene structure like faded edges, as seen in those rows.
>
> We also exploit more commonly used metrics to compare both methods. In particular, we additionally report quantitative metrics computed specifically for the stylization results shown in the second and third rows. For the 2D method (StyTr2), which cannot synthesize novel views, we used ground-truth novel views as input. This actually gives StyTr2 an advantage and makes the comparison with the 3D method conservative.
>
> Table r4: The ArtFID and Histogram results for the second and third row.
> |             | **Styl3R** | **StyTr2 (2D)** |
> |:------------|:----------:|:-------------:|
> | ArtFID ↓ (2nd)|   **29.42**    |     29.91    |
> | ArtFID ↓ (3rd)|   **26.94**    |     28.73     |
> | Histogram ↓(2nd)|   **0.214**    |     0.216     |
> | Histogram ↓(3rd)|   **0.148**    |     0.231     |
>
> As shown above, our method performs slightly better than the 2D baseline in both ArtFID and Histogram metrics.
>
> ---
>
> > **Q: Why are two DPT heads used—one for predicting the center position and another for other attributes? What is the motivation behind this design choice?**
>
> We leverage the pretrained weights of MASt3R, thus this separation can make us effectively use the pretrained center head from MASt3R without the need of training a new head from scratch.

---

### Official Review · Reviewer_KFYQ · 2025-06-29

**Clarity:** 3
**Significance:** 1
**Originality:** 2
**Rating:** 4
**Confidence:** 5

**Summary:**

The paper combines feed-forward reconstruction networks with stylization of the generated output scenes. At the core, there is a dual-branch architectures that decouples appearance and structure modeling in a joint network.

2-8 input images -> stylized 3dgs as output.

**Questions:**

What is the motivation of combining reconstruction and stylization in a joint architecture? Why not do first reconstruction once, and then independently stylize the respective gaussians? This would have certain advantages in not having to share the capacity of the reconstruction / stylization network.

This makes it feel a bit like an academic task for a paper but practically speaking I would not expect anyone re-reconstruct everything when a new style query comes in. As outlined, I would rather expect a stylization method that operates on a given reconstruction (that would also allow to work on more complete 3DGS outputs). A discussion here would be great.

**Ethical Concerns:**

["NO or VERY MINOR ethics concerns only"]

**Final Justification:**

I appreciate the authors addressing several of my concerns in the rebuttal and following discussions. I'm still somewhat on the fence, but I'm now leaning mildly positive (borderline accept).

However, in case the paper gets accepted, I would ask the authors to incorporate the promised experiments and discussions as outlined.

**Limitations:**

There is a short discussion at the end of the conclusion, but it's not very comprehensive and definitely is not a serious attempt to discuss the method's limitations. Generally, it makes it feel a bit that there is not a clear understanding of the bigger picture as the practical limitations would be very obvious.

**Paper Formatting Concerns:**

no concerns

**Quality:**

3

**Strengths And Weaknesses:**

Strengths
- reasonable results for the limited view point changes
- seems the first method to combine feed-forward reconstruction with stylization

Weaknesses
- inherits all the weaknesses from early feed-forward reconstruction. Essentially, the inputs are limited to a very small number of views which leads to rather small reconstruction. For instance, in the video there is barely any view point change which makes it look rather than view interpolation than any real reconstruction.
- the pipeline is a bit incremental; essentially, take feed-forward reconstruction techniques and include a style component.

-> Overall, I'm a bit on the fence, the task is a bit artificial (see below) and the results are limited given the small number of views. On the other hand the results are not too bad, and the inference speed is quite fast.

---

> ### Author Rebuttal · Authors · 2025-07-30
>
> We sincerely thank the reviewer for recognizing the originality of our approach in combining feed-forward reconstruction with stylization, and for acknowledging the reasonable results it achieves. We have addressed the reviewer’s comments in detail below, and would be happy to clarify further if needed.
>
> > **W1: inherits all the weaknesses from early feed-forward reconstruction. Essentially, the inputs are limited to a very small number of views which leads to rather small reconstruction. For instance, in the video there is barely any view point change which makes it look rather than view interpolation than any real reconstruction.**
>
> We agree that reconstruction from sparse, unposed views remains a challenging open problem, and Styl3R inevitably inherits some of the limitations of earlier feed-forward pipelines such as MASt3R. However, Styl3R is the first to enable stylization of arbitrary 3D scenes with arbitrary styles from sparse, unposed input views. This problem setting has not been explored by prior 3D stylization methods. We imagine Styl3R would inspire future follow-up works to handle more input views, as how Dust3R/MaSt3R inspires VGGT (CVPR 2025), which demonstrates promising 3D reconstruction performance even with many input images.
>
> On the other hand, in our humble opinion, the viewpoint changes are actually not that small from the video. As shown between 6–30 seconds in the supplementary video, the input views have limited spatial overlap, and the time interval between the two views is around 5 seconds, indicating substantial view variation. We further tested Styl3R with more than 8 input views, and in some scenes, it successfully handles view changes up to 180 degrees. We also additionally re-train the model with random 2 to 6 view inputs like VGGT, Styl3R achieves even broader viewpoint coverage. Due to the rebuttal policy, we could not post the qualitative results here, we would include the results in the final version.
>
> As for "view interpolation", Styl3R is not performing view interpolation. The model generates explicit 3D Gaussians, and the outputs are rendered from novel camera viewpoints that are unseen during training, demonstrating 3D reconstruction capability.
>
> ---
>
> > **W2: the pipeline is a bit incremental; essentially, take feed-forward reconstruction techniques and include a style component.**
>
> We acknowledge that Styl3R builds upon the architectures of DUSt3R and MASt3R for the structure branch. However, these models were not designed for or applied to the task of 3D stylization. As noted by reviewers *EHSJ* and *4c71*, Styl3R tackles the novel and underexplored problem of 3D scene style transfer.
>
> Moreover, our appearance branch is entirely novel, designed to support seamless switching between stylized and original appearances, and is trained from scratch. This design enables arbitrary style-scene blending in a purely feed-forward manner, which, to our knowledge, has not been explored before in the context of 3D stylization.
>
> ---
>
> > **W3: Overall, I'm a bit on the fence, the task is a bit artificial (see below) and the results are limited given the small number of views. On the other hand the results are not too bad, and the inference speed is quite fast.**
>
> We respectfully disagree with the comment that "the task is a bit artificial." It has various application scenarios, even for some products:
> - Sparse-view 3D stylization is increasingly relevant for mobile content creation. For instance, Snap Inc. has explored 3D stylization in products like Snapchat, which supports short-form video, AR creation, and visual messaging. These are scenarios where real-time and casual stylization is essential.
> - Styl3R also has the potential to be extended to support text-conditioned 3D stylization, allowing it to generate stylized 3D scenes from user-provided text descriptions. Since our appearance encoder is image-trained, we adopt a two-stage approach as a tryout: using ChatGPT to generate a style image from the text prompt, which is then fed into Styl3R. This enables Styl3R to perform stylization from only text and sparse, unposed views, greatly enhancing flexibility and real-world applicability. It also aligns with recent advances in world models, such as *HunyuanWorld 1.0* [3], which also investigates the connection between language and 3D scene generation or stylization.
> - Since prior test-time optimization based methods require long time to get the stylized 3D content, Styl3R could also serve as a preview approach for the consumers to quickly try different style images which would best fit their tastes. Then they could potentially exploit prior methods (which are time consuming) to get an even larger stylized 3D scene.
>
> This suggests that Styl3R contributes meaningfully to an emerging and impactful research direction.
>
> [3] Tencent HunyuanWorld Team. "HunyuanWorld 1.0: Generating Immersive, Explorable, and Interactive 3D Worlds from Words or Pixels." Technical Report, ArXiv, July 2025.
>
> > **Q1: What is the motivation of combining reconstruction and stylization in a joint architecture? Why not do first reconstruction once, and then independently stylize the respective gaussians? This would have certain advantages in not having to share the capacity of the reconstruction / stylization network.**
>
> We choose to combine reconstruction and stylization in a joint, end-to-end architecture to better support interactive applications, such as mobile photography and AR filters, where on-the-fly switching of digital effects is often required.
>
> In these scenarios, casual users may simply hold up their phone with modest movement and wish to apply different visual styles to the 3D scene in real time. Styl3R is currently **the only 3D stylization method** capable of handling this efficiently, as all prior methods require dense inputs and slow optimization procedures. This advantage in real-world interactivity was also recognized by *Reviewer 4c71*.
>
> Regarding network design, since stylization operates directly on 3D Gaussians, allowing the stylization module to be aware of scene structure helps maintain multi-view consistency and spatial coherence. Therefore, sharing certain components between the reconstruction and stylization branches is not only efficient, but also a principled architectural choice.
>
> ---
>
> > **Q2: This makes it feel a bit like an academic task for a paper but practically speaking I would not expect anyone re-reconstruct everything when a new style query comes in. As outlined, I would rather expect a stylization method that operates on a given reconstruction (that would also allow to work on more complete 3DGS outputs). A discussion here would be great.**
>
> If the scene images do not change and only the style image changes (i.e. re-stylization), Styl3R could do the re-stylization without reconstructing the 3D scene again.  For a given scene, we perform a single forward pass through the structure branch and cache the generated 3D Gaussians without appearance attributes. Then, for different style queries, we only need to run the appearance branch, which efficiently updates the stylized appearance while keeping the scene geometry fixed. This also highlights the motivation/advantage behind our dual-branch design, which cleanly separates structure and appearance for efficient re-stylization.
>
> We agree that a stylization method that operates directly on a complete precomputed 3D Gaussians also has great values for certain scenarios. However, Styl3R has its own advantages for some other application scenarios as described in previous section, especially for time-constrained use cases (e.g. quick 3D content stylization preview etc.). In our humble opinion, we think both problems targets for different applications, they could even complement with each other for certain scenarios.

---

> > ### Comment · Reviewer_KFYQ · 2025-08-04
> >
> > Thanks for the clarifications in the rebuttal - very much appreciated.
> >
> > I'd like to understand better some of the responses, in particular to the real-world applications:
> > - what is the connection to the Snap filters. Could you provide a reference here and how they explore and/or are looking for 3D stylization? At the moment this sounds pretty broad.
> > - the authors mention HunyuanWorld that recently appeared. To my understanding, they generate panorama images with several layers of depth estimates and then use a version of multi-plane image rendering. How does this relate to a feed-forward 3DGS + stylization approach?
> > - Could you please again re-emphasize the advantage over 3DGS + stylization in a single pass vs 3DGS first, and then the stylization in a subsequent steps. I can think of some but I would feel more comfortable if this tradeoff is addressed in an honest pro and con discussion.
> > - Re: limited few points: could you provide some detail how to run the model with more than 8 views? Further, I am somewhat concerned with the claim "it successfully handles view changes up to 180 degrees". To my understanding, most LRM/feed-forward-based 3DGS methods w/o stylization struggle with this and it's a fundamental limitation of these type of architectures. Why would it work better here?

---

> ### Author Response · Authors · 2025-08-06
> **Replying to Official Comment by Reviewer KFYQ -- Part1**
>
> Thank you very much for your valuable follow-up questions. Please find our responses as follows:
> > **Q1: what is the connection to the Snap filters. Could you provide a reference here and how they explore and/or are looking for 3D stylization? At the moment this sounds pretty broad.**
>
> Since we are not allowed to use links according to the review policy, you can try to Google search "Stylized Digital Art Filter by Snapchat". The first result should present a stylized lady as well as stylized background. We are unsure on which technique they exploit to achieve the effect since it is a commercial product. However, if we look into the background as the video plays, the views are not that consistent. For this kind of application scenarios, efficiency is very important for the user experience, which motivates our work. Styl3R is able to deliver more view consistent stylized video by rendering from the predicted stylized 3D Gaussians in a very efficient way. Although Styl3R still could not handle dynamic scenes, we believe it could inspire future works to address this limitation.
>
> ---
>
> > **Q2: the authors mention HunyuanWorld that recently appeared. To my understanding, they generate panorama images with several layers of depth estimates and then use a version of multi-plane image rendering. How does this relate to a feed-forward 3DGS + stylization approach?**
>
> We appreciate the reviewer’s thoughtful follow-up. While HunyuanWorld and Styl3R differ significantly, HunyuanWorld has the capability for stylized 3D scene generation from language inputs. Our reference to HunyuanWorld is not intended to imply architectural similarity, but rather to highlight the emerging trend of linking language/image inputs with stylized 3D scene creation. Our prototype extension (text → style image → Styl3R) demonstrates a lightweight alternative in this trend, offering interactive stylization capabilities from multi-view images.
>
> ---
>
> > **Q3: Could you please again re-emphasize the advantage over 3DGS + stylization in a single pass vs 3DGS first, and then the stylization in a subsequent steps. I can think of some but I would feel more comfortable if this tradeoff is addressed in an honest pro and con discussion.**
>
> Comparison: Two-Stage vs. Single-Pass 3DGS Stylization
>
> Two-Stage Approach: 3DGS + Stylization in Subsequent Steps
>
> Pros:
> - High-fidelity geometry: The 3DGS model is optimized directly to fit the target scene from dense image inputs, resulting in more accurate geometry and detailed structure prior to stylization.
> - Scalable to large scenes: With sufficient optimization time and resources, this approach can be extended to larger-scale environments.
>
> Cons:
> - No instant stylization support: There is currently no method that directly takes a trained 3DGS and outputs a stylized 3DGS without additional test-time optimization. Even the closest work (e.g., three-stage training approach: StyleGaussian) requires post-processing steps like VGG feature embedding and AdaIN.
> - Efficiency: The optimization-heavy pipeline is unsuitable for interactive or time-sensitive applications such as AR or mobile capture.
> - Requires expert setup: Users need to collect dense, accurately posed input images to fit 3DGS properly, limiting accessibility for casual creators.
>
> Single-Pass Approach: 3DGS + Stylization in a single pass
>
> Pros:
> - Fast inference: The model runs in a single forward pass with no test-time optimization, enabling nearly real-time 3D stylization—ideal for interactive or mobile scenarios.
> - User-friendly: The pipeline accepts a small set of casual images and produces a stylized 3D scene in an end-to-end manner, making it accessible to non-experts.
>
> Cons:
> - Data-hungry: The training stage requires large dataset to enable generalization across a wide range of scene types and artistic styles.
> - Lower reconstruction quality: Feed-forward models usually prioritize speed and generalization over precise geometric fidelity.
> - Limited scalability to large-scale scenes: Current implementations may face memory constraints (e.g., VRAM bottlenecks) and require architectural adjustments or more training data to support larger-scale environments.
>
> Inspired by a recent work from this year's CVPR, i.e. VGGT [1], which has demonstrated impressive reconstruction quality with a large number of input views, we believe our work could potentially inspire follow up works along this direction to address the last two limitations.
>
> [1] Wang, Jianyuan, et al. "VGGT: Visual Geometry Grounded Transformer." CVPR. 2025.

---

> ### Author Response · Authors · 2025-08-06
> **Replying to Official Comment by Reviewer KFYQ -- Part2**
>
> > **Q4: Re: limited few points: could you provide some detail how to run the model with more than 8 views? Further, I am somewhat concerned with the claim "it successfully handles view changes up to 180 degrees". To my understanding, most LRM/feed-forward-based 3DGS methods w/o stylization struggle with this and it's a fundamental limitation of these type of architectures. Why would it work better here?**
>
> Technically, our model can handle more input views (e.g., more than 10), as long as sufficient GPU memory is available.
>
> - In the structure branch, each view’s tokens attend to the concatenated tokens from all other views via cross-attention in the decoder, enabling the model to infer the structural attributes of Gaussians with multi-view consistency.
> - In the appearance branch, tokens from all views are concatenated and then do cross-attention to the style tokens.
>
> This design is likewise not inherently limited by the number of views during inference, but could increase the memory usage, especially for the structure branch. A more efficient attention mechanism could be exploited in future in order to run the model with a large number of inputs.
>
> To explore the model's ability to cover larger viewpoint ranges, we conducted experiments on DL3DV, which contains scenes captured with circular camera motions around medium-scale objects (e.g., statues, flowerbeds). For testing, we evenly selected 12 views spanning 180 degrees from the circular path. For some scenes of this scale, the model produces reasonable results over a 180-degree range, with slight degradation though. Since current model is trained with only 4 views, it holds strong potential for improved performance when trained with more input views.
>
> Some recent LRM- or feed-forward-based 3DGS methods demonstrate similar generalization to wide view ranges:
> - LVSM [2] achieves 360° object-level reconstruction with just 4 input views.
> - DepthSplat [3] achieves near-360° large-scale scene reconstruction with 12 views.
>
> For our model, the key limiting factor in covering wide baselines lies in the pixel-aligned Gaussian prediction, i.e. whether Gaussians derived from different views can be properly aligned in 3D space. When the view differences are not too significant and alignment remains accurate, increasing the number of input views directly extends the angular coverage of the reconstruction without requiring architectural changes.
>
> [2] Jin, Haian, et al. "LVSM: A Large View Synthesis Model with Minimal 3D Inductive Bias." ICLR. 2025.
>
> [3] Xu, Haofei, et al. "DepthSplat: Connecting Gaussian Splatting and Depth." CVPR. 2025.

---

> > ### Author Response · Authors · 2025-08-07
> >
> > Dear Reviewer KFYQ:
> >
> > We hope our response has addressed your questions. As the discussion phase is coming to a close, we are looking forward to your feedback and would like to know if you have any remaining concerns we can address. We are grateful if you find our revisions satisfactory and consider raising your score for our paper.
> >
> > Thank you once again for the time and effort you have dedicated to reviewing our paper.
> >
> > Best regards
> >
> > Styl3R Authors

---

> > > ### Comment · Reviewer_KFYQ · 2025-08-07
> > >
> > > Thank you for the detailed response - really appreciated.
> > >
> > > Re Q1/2: obviously research papers do not have to be straight ready to be a product, but the motivation based on Snap Lenses and HunyuanWorld is *really* far fetched - both technically as well as application wise. I don't find this argumentation convincing, in particular since any other 3D reconstruction or stylization approach (which there are plenty) could claim the same thing.
> > >
> > > Re Q3: that's a good response, and this discussion should really go into the paper as it would motivate why this specific design choices was made. The main pro argument of the approach being that it's faster to do it one pipeline. At the same time, I feel one main downside is that keeping the style consistent across multiple reconstructions is challenging. Such an incremental update / re-reconstructing would be the main application that method would need to do in any real time context where speed matters.
> > >
> > > Re Q4: the issue with the reply here is that the complexity of the view point changes is entirely dependent on the underlying reconstruction method, which is not where the authors innovate. In principle this fine but I don't think the authors should make claims what is not shown in the results (and what the respective base models can't support either).
> > >
> > > At this point, I don't have any further questions.

---

> ### Author Response · Authors · 2025-08-08
>
> Thank you very much for your response and constructive suggestions.
>
> **Re-Re Q1/Q2:**
> We would like to further clarify that this project is primarily curiosity-driven research originally, aimed at exploring whether existing 3D stylization methods can move beyond slow, optimization-heavy pipelines.
>
> The purpose to take Snap Lenses as an example is simply to demonstrate that Styl3R or its future follow-up works could potentially be applied to those application scenarios, which require instant (view consistent) stylization for better user experience. It could also be applied to other scenarios, such as quick 3D stylization preview as described in the original response.
>
> **Re-Re Q3:**
> Thank you for your constructive suggestions, we will include the discussion on the pros and cons to clarify our design choices.
>
> In our current work, we mainly focus on how to maintain consistency within a single feed-forward reconstruction. Keeping the style consistent across multiple different reconstructions is indeed what Styl3R currently struggles. In our humble opinion, it is actually not the focus of our current work.
>
> While Styl3R has limitations (e.g. limited number of input views), IMHO, the paper's contributions should outweigh its disadvantages to further advance the field of 3D stylization, by considering prior literatures.
>
> **Re-Re Q4:**
> Thanks for your suggestions. We would make claims more rigorous in the future.
>
> Thank you once again for your efforts in reviewing our paper, and providing so many constructive comments. We would incorporate these suggestions into our final paper.

---

### Official Review · Reviewer_BqGQ · 2025-07-02

**Clarity:** 3
**Significance:** 2
**Originality:** 2
**Rating:** 3
**Confidence:** 3

**Summary:**

This paper proposes a feed-forward network called Styl3R, which can achieve instant 3D stylization from sparse, unposed input views and a single reference style image. The network includes a structure branch and an appearance branch, and can generalize to arbitrary scenes and styles without test-time optimization. It outperforms existing baselines in zero-shot stylization quality and has a faster inference speed, making it suitable for real-world and interactive applications.

**Questions:**

Could the authors provide more examples of potential downstream applications? The style interpolation example is interesting, but more applications would strengthen the paper's impact.

**Ethical Concerns:**

["NO or VERY MINOR ethics concerns only"]

**Final Justification:**

After carefully reviewing the authors' rebuttal and subsequent discussions, I maintain my original assessment that the novelty of the paper is insufficient. While the authors have elaborated on their architectural designs (such as the dual-branch structure and attention-based fusion) and emphasized the "new problem setting" of instant 3D stylization with sparse, unposed inputs, these contributions still primarily reflect effective integration of existing reconstruction frameworks (e.g., MASt3R) and mature stylization techniques (e.g., content/style losses), rather than fundamental algorithmic innovations. The authors' responses do not adequately address my core concern regarding the limited novelty of the work. Therefore, I remain inclined to uphold the "Borderline reject" rating.

**Limitations:**

The primary concern is the limited novelty of the paper. Many of the techniques employed, such as the two-stage training and the use of content and style losses, were previously introduced in methods like StyleGaussian. The core contribution appears to be the adaptation of these existing stylization techniques to the MASt3R  framework.

**Paper Formatting Concerns:**

No formatting concerns

**Quality:**

3

**Strengths And Weaknesses:**

Strengths
1. The approach presented in the paper is direct and easy to follow.
2. The proposed model architecture is well-designed, and the experiments are comprehensive.

Weaknesses
1. The model's experimental results, both qualitative and quantitative, do not show a significant improvement over ARF.
2. The paper lacks a user study to subjectively evaluate the stylization quality.
3. The high speed of Styl3R is not attributed to a novel architecture proposed by the authors, but rather stems from the inherent speed of the base models (DUSt3R and MASt3R ). Therefore, this speed advantage does not appear to be a direct contribution of this work.
4. The effect shown in the ablation study for the Content Loss Layers (Figure 7) is not sufficiently convincing from the visual results alone. Could the authors provide quantitative experiments to better support this claim?

---

> ### Author Rebuttal · Authors · 2025-07-30
>
> We sincerely thank the reviewer for recognizing the *simplicity and effectiveness of our framework*, as well as the *thoroughness of our experimental evaluation*. We have addressed the reviewer’s concerns in detail below and would be happy to clarify further if needed.
> > **W1: The model's experimental results, both qualitative and quantitative, do not show a significant improvement over ARF.**
>
> While we appreciate the strength of ARF as a baseline, we emphasize that **Styl3R (ours)** and ARF operate under fundamentally different assumptions, as summarized in *Table 1* in the main paper:
> - ARF requires dense, posed multi-view images, and performs per-scene and per-style test-time optimization, taking approximately 10 minutes for scene fitting and 2 minutes for stylization. If we want to try different stylization images, ARF has to repeat the time-consuming optimization procedure;
> - Styl3R, on the other hand, operates in a fully feed-forward manner, requiring only unposed sparse-view images and a style image. It predicts stylized 3D Gaussians in only 0.15 seconds, without needing any optimization or camera supervision during inference stage. It makes Styl3R better suited for real-world and interactive scenarios;
> - Furthermore, from the results of the main paper (e.g. *Table 2*) as well as following newly conducted comparisons (*Table r1*, *Table r2* as shown below), Styl3R also consistently performs better than ARF. For example, in terms of consistency metric, Styl3R achieves 0.044 vs. 0.093 of ARF (*Table 2*); in terms of run time, Styl3R requires 0.147s vs. 12mins of ARF (*Table 2*).
>
> ---
>
> > **W2: The paper lacks a user study to subjectively evaluate the stylization quality.**
>
> Thank you for the valuable suggestion. To complement our quantitative evaluation, we conducted a user study to subjectively assess stylization quality with 20 3D style transfer experiments: 16 comparisons against baselines (13 from RE10K and 3 from Tanks and Temple) and 4 ablation comparisons on content loss layers. For the 16 baseline cases, participants viewed seven 256×256 images: a context image, a style image, and five method outputs (Styl3R, ARF, StyleGS, StyleRF, StyTr2) in a blind, randomized order, all rendered from the same target viewpoint. For the 4 ablation cases, each set included four images: context, style, and outputs from models trained with either h3 or h3+h4 content loss layers, also shown blindly and randomly.
>
> Participants selected the result they found to be the most harmonious combination of style and scene. The study was conducted online with volunteer participants. In total, we collected around 2,000 votes: 1,274 for RE10K, 294 for Tanks and Temple, and 392 for the ablation study.
>
> The stylized images shown for voting were novel views synthesized by 3D methods (Styl3R, ARF, StyleGS, StyleRF), rather than the training views. For the 2D method (StyTr2), which cannot synthesize novel views, we directly fed the ground-truth novel views as input — this provides an advantage to StyTr2, making the comparison against 3D methods conservative. The result is shown below:
>
> Table r1: The voting results for stylization comparision.
> | |Styl3R|ARF|StyleGS|StyleRF|*StyTr2 (2D)*|
> |:-|:-:|:-:|:-:|:-:|:-:|
> |Re10K votes (%)|**53.29**|14.13|6.67|1.81|24.10|
> |TnT votes (%)|**38.78**| 22.11| 11.56 |3.06|24.49|
>
> In addition to the user study, we also employed ArtFID [1], a metric known to align well with human judgment, as it jointly considers content preservation and style fidelity. We further used the RGB-uv histogram from HistoGAN [2] to evaluate color transfer performance. The result is shown below:
>
> Table r2: The ArtFID and Histogram metrics for stylization comparison.
> |Metric|Styl3R| ARF|StyleGS|StyleRF|*StyTr2 (2D)*|
> |:-|:-:|:-:|:-:|:-:|:-:|
> |ArtFID ↓ / Re10K|**35.12**|42.95|55.75|46.59|38.93|
> |ArtFID ↓ / TnT|**38.05**|48.98|55.03|64.81|39.24|
> |Histogram ↓ / Re10K|**0.230**|0.313 |0.465|0.507|0.241|
> |Histogram ↓ / TnT|**0.241**|0.259|0.379|0.422|0.247|
>
> ➤ *Summary:* Both user voting and quantitative metrics demonstrate that Styl3R achieves better stylization performance aligned with human judgment, even on out-of-domain datasets (*TnT*).
>
> [1] Wright, Matthias, and Björn Ommer. "ArtFID: Quantitative Evaluation of Neural Style Transfer." DAGM German Conference on Pattern Recognition, 2022.
>
> [2] Afifi, Mahmoud, Marcus A. Brubaker, and Michael S. Brown. "HistoGAN: Controlling Colors of GAN-generated and Real Images via Color Histograms." CVPR, 2021.
>
> ---
>
> > **W3: The high speed of Styl3R is not attributed to a novel architecture proposed by the authors, but rather stems from the inherent speed of the base models (DUSt3R and MASt3R ). Therefore, this speed advantage does not appear to be a direct contribution of this work.**
>
> We acknowledge that Styl3R builds upon the architectures of DUSt3R and MASt3R for the structure branch. However, these models were not designed for or applied to the task of 3D stylization. As also agreed by reviewers *EHSJ* and *4c71*, Styl3R tackles the novel and underexplored problem of 3D scene style transfer.
>
> Moreover, our appearance branch is entirely original, designed to enable seamless switching between stylized and original appearances, and is trained from scratch. This demonstrates that arbitrary style-scene blending can be achieved in a feed-forward manner, which has not been explored previously.
>
> ---
>
> > **W4: The effect shown in the ablation study for the Content Loss Layers (Figure 7) is not sufficiently convincing from the visual results alone. Could the authors provide quantitative experiments to better support this claim?**
>
> As noted in *W2*, we conducted a user study (392 votes) and computed ArtFID and Histogram metrics to compare the h3 layer and h3+h4 combination. Results are shown below:
>
> Table r3: The ArtFID, Histogram and voting results for h3 layer and h3+h4 layer ablations.
> ||ArtFID ↓|Histogram ↓|Votes (%) |
> |:-|:-:|:-:|:-:|
> |h3 layer|42.01|0.277|35.20|
> |h3+h4 layer|**35.83**|**0.239**|**64.80**|
>
> ➤ *Summary:* All three metrics confirm that the adopted h3+h4 configuration yields better performance.
>
> ---
>
> > **Q: Could the authors provide more examples of potential downstream applications? The style interpolation example is interesting, but more applications would strengthen the paper's impact.**
>
> We explore text-conditioned 3D stylization, enabling Styl3R to stylize 3D scenes based on user-provided text descriptions. Because our current appearance encoder is trained solely on images, we adopt a two-stage approach to bridge the gap: first, we use ChatGPT to generate a style image from a text prompt, and then feed the generated image into Styl3R as the style input. In our evaluations, this enables Styl3R to perform 3D stylization using only text descriptions and sparse, unposed views, greatly enhancing its flexibility and real-world applicability. Moreover, extending the encoder to directly support text input is technically feasible and represents a promising direction for future work.
>
> This line of exploration aligns with recent advances in world models, such as *HunyuanWorld 1.0* [3], which also investigates the connection between language and 3D scene stylization.
>
> [3] Tencent HunyuanWorld Team. "HunyuanWorld 1.0: Generating Immersive, Explorable, and Interactive 3D Worlds from Words or Pixels." Technical Report, ArXiv, July 2025.
>
> ---
>
> > **Limitation: The primary concern is the limited novelty of the paper. Many of the techniques employed, such as the two-stage training and the use of content and style losses, were previously introduced in methods like StyleGaussian. The core contribution appears to be the adaptation of these existing stylization techniques to the MASt3R framework.**
>
> We appreciate the reviewer’s concern and agree that some components, such as content/style losses and multi-stage training, have been explored in prior works like StyleGaussian. However, Styl3R differs significantly in both problem formulation and technical design, as outlined below:
>
> - Problem setting is novel: In our humble opinion, Styl3R is the first to address the problem of 3D scene style transfer in a feed-forward manner, which introduces a novel paradigm for 3D scene stylization problem. It is also agreed by reviewer EHSJ: "*The paper addresses a novel problem of 3D scene style transfor using sparse input, which to the best of my knowledge, has not been explored before... The paper presents the first framework that applies feedforward networks to 3D scene stylization.*"
>
> - The approach is novel: Styl3R introduces a token-level stylization module using transformer-based cross-attention, directly blending multi-view content and style features for globally coherent results. It is also agreed by reviewer 4c71 & EHSJ: "*​Novel Architecture: This is the first work that extends feed-forward 3D reconstruction architecture, such as Dust3R, NoPoSplat, to 3D stylization. ​Transformer-based stylization decoder​ ensures multi-view consistency by blending global style and content features.*" and "*The approach is both novel and interesting.*"
>
> - Good performance: Compared to prior works on 3D scene stylization, Styl3R also achieves higher stylization quality, better consistency, and improved efficiency.
>
> ➤ *Summary:* Although Styl3R builds upon common stylization losses, its problem setting, technical innovations, and good empirical performance, make it a good step forward from prior test-time optimization based methods like StyleGaussian.

---

> > ### Comment · Reviewer_BqGQ · 2025-08-04
> > **Response**
> >
> > My core argument is that Styl3R skillfully combines an existing efficient reconstruction framework (similar to MASt3R) with a set of mature stylization techniques (similar to StyleGaussian, such as VGG loss and two-stage training).
> > While the authors emphasize in their rebuttal that they have addressed a new problem setting (instant stylization with sparse, pose-free inputs), this is more akin to a highly successful "system integration" rather than a "fundamental innovation" at the algorithmic level.

---

> ### Author Response · Authors · 2025-08-05
> **Replying to Official Comment by Reviewer BqGQ**
>
> We greatly appreciate the reviewer’s thoughtful assessment. Sorry for the confusion and we would make it more clear. We acknowledge that Styl3R builds upon prior advances in reconstruction (e.g., MASt3R) and uses standard stylization losses commonly adopted in style transfer literature (though not introduced by StyleGaussian). However, we respectfully argue that the **novelty of Styl3R lies not merely in integration**, but also in its **architectural innovations specifically tailored to a more challenging problem setting**. We elaborate on these contributions below.
>
> ---
>
> Differences with MASt3R and StyleGaussian:
>
> ---
>
> 1. Dual-branch Design:
>
> Unlike prior methods (e.g., MASt3R), which couple structure and appearance modeling within a single stream, Styl3R introduces a novel dual-branch architecture that explicitly disentangles structural estimation from appearance modeling. This separation ensures that stylization operations do not interfere with the reconstruction of 3D geometry.
>
> ---
>
> 2. Token-level Features:
>
> StyleGaussian adopts a three-stage training strategy:
>
> &nbsp;&nbsp;&nbsp;&nbsp;1) Per-scene RGB optimization: Standard 3D Gaussians are trained from multi-view images using known poses;
>
> &nbsp;&nbsp;&nbsp;&nbsp;2) VGG feature embedment: VGG features extracted from the RGB images are embedded into the Gaussians;
>
> &nbsp;&nbsp;&nbsp;&nbsp;3) Stylization decoder training: A per-scene decoder is trained using content and style losses to stylize the Gaussians transformed via AdaIN in the VGG feature space.
>
> However, VGG features are inherently local due to the convolutional nature of CNNs, and thus lack the capacity to capture long-range semantic consistency. Furthermore, the stylization decoder in StyleGaussian explicitly operates on k-nearest neighboring Gaussians, which limits its stylization expressiveness by relying solely on local feature aggregation.
>
> As a result, StyleGaussian often suffers from structural distortions and loss of fine details, as demonstrated in our experiments (see Table 2 and Figure 3).
>
> In contrast, Styl3R removes the need for intermediate VGG-Gaussian representations and directly predicts stylized Gaussians from token-level features. These features are produced by a transformer-based stylization module, which globally blends multi-view content and style information—leading to more coherent and consistent 3D stylization results.
>
> ---
>
> 3. Attention-based Content Style Fusion:
>
> To stylize Gaussians, StyleGaussian employs AdaIN [1] to align the mean and variance of VGG features between the reconstructed Gaussians and the style image. However, this heuristic, statistic-based approach is suboptimal and lacks semantic awareness, making it difficult to preserve fine details and local fidelity. In practice, we observe that AdaIN often leads to over-smoothing and under-stylization, particularly in complex regions (see Fig. 3 and Table 2).
>
> In contrast, Styl3R leverages an attention-based mechanism that automatically learns to blend multi-view scene content with the provided style, while preserving cross-view consistency. Rather than relying on handcrafted feature alignment like AdaIN, Styl3R’s attention-based fusion dynamically models content-style correspondences across views.
>
> This results in globally coherent, structurally aligned stylization. **To the best of our knowledge, this token-level fusion strategy is the first of its kind in the 3D stylization domain**, and is a key enabler of Styl3R’s zero-shot generalization without test-time optimization.
>
> [1] Huang, Xun, and Serge Belongie. "Arbitrary style transfer in real-time with adaptive instance normalization." ICCV. 2017.

---

> > ### Author Response · Authors · 2025-08-07
> >
> > Dear Reviewer BqGQ:
> >
> > We hope our response has addressed your questions. As the discussion phase is coming to a close, we are looking forward to your feedback and would like to know if you have any remaining concerns we can address. We are grateful if you find our revisions satisfactory and consider raising your score for our paper.
> >
> > Thank you once again for the time and effort you have dedicated to reviewing our paper.
> >
> > Best regards
> >
> > Styl3R Authors

---

### Decision · Program_Chairs · 2025-09-17

**Decision:**

Accept (poster)

**Comment:**

Styl3R is a feed-forward network for instant 3D stylization from sparse, unposed input views and a single style image.
The method employs a dual-branch architecture to disentangle structure and appearance, with a transformer-based decoder and identity loss to ensure multi-view consistency. It achieves fast inference, requires no test-time optimization and demonstrates strong stylization quality and consistency compared to prior baselines.

The work demonstrates clear improvements over existing 3D stylization methods, especially in zero-shot generalization and efficiency, making it suitable for interactive applications. Reviewers found the problem setting novel and impactful, and appreciate the network architecture and strong empirical results.

Concerns remain around technical novelty (integration of existing reconstruction + stylization components), limited viewpoint diversity due to sparse inputs, and dependency on base model speed.

Nonetheless, the paper proposes a practical pipeline that addresses a novel task, outperforms reasonable baselines, and is likely to influence future 3D stylization research. After rebuttal, reviewers acknowledged that their concerns were largely addressed, and the consensus shifted positively.

Hence, I recommend accepting the submission as a poster.